# Moving Surface Boundary-Layer Control on the Wake of Flow around a Square Cylinder

Te Song [1], Xin Liu [2] and Feng Xu [2,*]

1 Engineering Management Department, Guangzhou Communications Investment Group Co., Ltd., Guangzhou 510290, China; estimos@163.com
2 School of Civil and Environmental Engineering, Harbin Institute of Technology (Shenzhen), Shenzhen 518055, China; 21b954008@stu.hit.edu.cn
* Correspondence: xufenghit@hit.edu.cn

**Abstract:** In this paper, the entire process of the flow around a fixed square cylinder and the moving surface boundary-layer control (MSBC) at a low Reynolds number was numerically simulated. Two small rotating circular cylinders were located in each of the two rear corners of the square cylinder, respectively, to transfer momentum into the near wake behind the square cylinder. The rotations of the two circular cylinders were realized via dynamic mesh technology, when the two-dimensional incompressible Navier–Stokes equations for the flow around the square cylinder were solved. We analyzed the effects of different rotation directions, wind angles $\theta$, and velocity ratios $k$ (the ratio of the tangential velocity of the rotating cylinder to the incoming flow velocity) on the wake of flow around a square cylinder to evaluate the control effectiveness of the MSBC method. In the present work, the aerodynamic forces, the pressure distributions, and the wake patterns of the square cylinder are discussed in detail. The results show that the high suction areas near the surfaces of the rotating cylinders can delay or prevent the separation of the shear layer, reduce the wake width, achieve drag reduction, and eliminate the alternating vortex shedding. For a wind angle of $0°$, the inward rotation of the small circular cylinders is the optimal arrangement to manipulate the wake vortex street behind the square cylinder, and $k = 2$ is the optimal velocity ratio between the control effectiveness and external energy consumption.

**Keywords:** wake control; drag reduction; MSBC; square cylinder; numerical simulation

## 1. Introduction

The development of complex structures has led to the construction of structures with reduced stiffness, such as large-span bridges and high-rise buildings, and wind-induced forces have become the main controlling factors in the design of these types of structures. The suppression of vortex shedding can lead to the lessening of the unsteady forces acting on the bluff bodies, thus resulting in less wind-induced vibrations. In terms of the flow field, the flow control changes the fluid-induced loads by altering the turbulence structures and the flow characteristics, whereupon the purpose of suppressing structural vibrations is achieved. The flow control may be accomplished by manipulating the boundary-layer separation or the structure of the shear layer in the wake.

A passive flow control does not consume external energy to realize its control purpose [1–4]. Laima et al. [5] and Chen et al. [6] studied the turbulence vortex structures, vortex-induced vibration characteristics, and passive flow control methods of the twin-box-girder bridge section model through wind tunnel tests. From the perspective of vortex dynamics, Gao et al. [7] presented a selective review of recent progress on the mechanism of VIV (vortex-induced vibration) occurred in long-span bridges and proposed several passive and active flow control methods to manipulate the surrounding flow patterns around the girder. Li et al. [8] and Chen et al. [9] studied the wind-induced vibration characteristics

and wind–rain excitation mechanism of cylindrical cable structure by wind tunnel test and numerical simulation, which laid a foundation for the development of new flow control methods. To realize the control purpose, an active flow control channels the external energy into the flow field and injects the proper perturbation so that it may interact with the inner mode of the flow. Unlike the wall vibration, the bubble method, injection and suction [10–12], or the traveling wave wall [13–15], the moving surface boundary-layer control (MSBC) method is another type of active flow control.

The passive wake control behind a circular cylinder in a uniform flow for a Reynolds number (hereafter denoted as Re) ranging from 80 to 300 has been studied by Kubo et al. [16]. In their study, the vortex street behind the main cylinder still existed; however, the fluctuating lift and the form drag on the main cylinder significantly and monotonously decreased when Re increased from 80 to 300. Hwang and Yang [17] placed two splitter plates with the same length as the cylinder diameter along the horizontal centerline on the upstream and downstream of the cylinder and analyzed the aerodynamic forces and vortex shedding modes of the cylinder by adjusting the distances between the two plates and the cylinder. The study found that the maximum value of drag reduction was achieved 38.6% at a certain set of gap ratios. Assi et al. [18] investigated the suppression of cross-flow and in-line vortex-induced vibrations of a circular cylinder with a low combined mass-and-damping parameter of up to 0.014 by using two-dimensional control plates. The results showed that the maximum drag reduction was approximately 38% and occurred when using parallel plates. Koca and Genç [19,20] revealed the relationship between aerodynamic performance and vortex shedding from suction surface and wake of different types of wind turbine blades at a low Reynolds number by using the wind tunnel test, which provided a theoretical basis for further flow control research. In order to improve the aerodynamic performance and power output of wind turbine blades, Genç et al. [21,22] and Koca et al. [23] achieved the purpose of suppressing or eliminating laminar separation bubbles and reducing aerodynamic force by changing the surface roughness and arranging flexible membrane material at appropriate positions on the suction surface or on both suction and pressure surfaces of wind turbine blades. Bayramoğlu and Genç [24] replaced the flexible membrane material with piezoelectric material, which can achieve the dual purpose of flow control and energy harvesting. Malekzadeh and Sohankar [25] used a numerical simulation method to study the reduction effect of the aerodynamic forces and heat transfer of a control plate on a square cylinder at Re $= 50 - 200$ based on the width of the square cylinder ($W$). This study shows that the optimum position and width for the control plate are a distance of $3W$ away from the cylinder and a width of $0.5W$ at Re $= 160$, where the cylinder drag and the total drag (cylinder and plate) have a reduction of 86% and 37%, respectively, and the rms lift and drag coefficients on the cylinder have a 92% and 90% reduction compared to that of an isolated cylinder. Blowing and suction are also an effective way to control the wake of bluff bodies. Turhal and Çuhadaroğlu [26] investigated some aerodynamic parameters of the flow around perforated-surface square, horizontal, and diagonal cylinders at three different Reynolds numbers in a wind tunnel. The experimental results showed that the surface injections through the top-rear, rear, and all surfaces of a diagonal square cylinder reduce the drag coefficient for all the Reynolds numbers, while the injection through all surfaces only reduces the drag coefficient of a horizontal square cylinder. Sohankar et al. [27] numerically investigated the effects of uniform suction and blowing through the surfaces of a square cylinder on the vortex shedding, wake flow, and heat transfer at low Reynolds numbers. The results show that the lift and drag fluctuations for the optimum configuration decay, and the maximum reduction on the drag force are 61%, 67%, and 72% for Re = 70, 100, 150, and respectively. Chen et al. [28] and Gao et al. [29] realized the passive jet flow control by attaching hollow opening pipes on the cylinder surface and opening a slit in the middle of the cylinder section. Both methods have achieved the purpose of suppressing or delaying the alternating shedding vortices in the wake of the cylinder.

Based on the wind tunnel tests, the theoretical prediction, and the flow visualization, Munshi et al. [30] applied the MSBC to two-dimensional rectangular prisms to achieve drag reduction and suppression of flow-induced vibrations. The results indicated that the existence of the MSBC effectively eliminated the vortex resonance and galloping-type instabilities and that the maximum momentum injection corresponding to a velocity ratio of 3 was sufficient to delay boundary-layer separation and to achieve a minimum drag condition. On the basis of the existing research methods, Munshi et al. [31] conducted numerical simulations and analyzed the drag of two-dimensional flat plates and rectangular prisms with MSBC. Both the numerical and the flow visualization results showed narrowing of the wake as the momentum injection increased, leading to eventual suppression of the vortex shedding. Kubo et al. [32] studied the suppression of the aerodynamic response of square-sectioned tall structures with MSBC. The results of their study clearly showed that the boundary-layer control through the use of rotating cylinders at the leading edges of a two-dimensional square prism were effective in suppressing the vortex-excited and galloping oscillations. The rotor length and its position on the structure were the main factors affecting the effectiveness of MSBC. Kubo et al. [33] conducted extensive wind tunnel experiments to study the role of the MSBC in the suppression of torsional flutter of a shallow rectangular prism, which was one of the typical fundamental configurations of bridge deck sections. The results that they obtained from measurements of surface pressure distributions and aeroelastic vibration responses showed that the proposed boundary-layer control method through momentum injection was effective in suppressing the torsional flutter of a shallow rectangular prism. Modi and Deshpande [34] conducted wind tunnel tests to study the aerodynamics of a cubic structure in the presence of MSBC, using rotating circular cylindrical elements at the two adjacent vertical edges. The results revealed the significant effect of the MSBC, which changes the height of the boundary-layer separation and reattachment, thus affecting the fluid dynamical parameters.

The moving surface boundary-layer control (MSBC) is also called angular momentum injection. Through numerical simulation, Patnaik and Wei [35] proposed an angular momentum injection strategy to control the wake dynamics and turbulence behind a fixed square cylinder at Re = 200. This study observed a new zone formation of a recirculation-free zone (RFZ) behind the cylinder at $\zeta = 1.25$. With the increase of angular momentum injection, the zones of absolute and convective instabilities (AIZ and CIZ) are diminished, and the recirculation-free zone is enhanced. Finally, the wake turbulence is completely suppressed at $\zeta = 1.5$. Muddada and Patnaik [36] compared the wake characteristics of a flow past an isolated circular cylinder, with momentum injection in the subcritical Re range. The numerical flow visualization through streaklines and streamlines clearly demonstrated the effectiveness of the rotating-type actuators in modifying the turbulent vortex structures. As the ratio of the momentum input became higher, the flow tended to be smoother. Mittal [37] investigated the flow past a bluff body with two rotating control cylinders using two-dimensional numerical simulation. When the control cylinders rotated with a high speed, in a manner that the tip speed was five times the free-stream speed, the flow at Re = 100 reached a steady state. For Re = $10^4$, although the flow remained unsteady, the wake was much more organized and narrower compared to the one without control. Korkischko and Meneghini [38] verified the efficiency of the MSBC in suppressing vortex-induced vibrations (VIV); the flow control results in a mean drag of almost 60% compared to that of the plain cylinder, reduction at $U_c/U = 5$. The wake is highly organized and narrower compared to the one observed in cylinders without control.

In the present work, the entire process involving the flow around a fixed square cylinder and the MSBC was numerically simulated. The present numerical simulation started from the flow around a fixed square cylinder; when the alternating vortex shedding appeared and became stable, the MSBC was then activated to control the wake. Two small rotating circular cylinders were employed, which were located in the rear corners of the square cylinder, similar to Patnaik and Wei [35], to achieve momentum injection into the flow field. In this work, we mainly study the control effectiveness of the rotation direction

of the small circular cylinders, the wind angle, and the velocity ratio on the oscillating wake of the square cylinder. The aerodynamic force time histories and their statistics, the mean pressure distribution, and the vortex shedding pattern were studied to demonstrate the control effectiveness of the MSBC for different influencing factors and to explain the mechanism of the MSBC flow control method.

## 2. Numerical Calculation Model

### 2.1. Governing Equations of the Fluid Flow

For a two-dimensional incompressible flow, the governing equations, including the continuity equation and the Navier–Stokes equation, in the Cartesian coordinate system can be written as:

$$\frac{\partial u_i}{\partial x_i} = 0, \tag{1}$$

$$\frac{\partial u_i}{\partial t} + u_j \frac{\partial u_i}{\partial x_j} = -\frac{1}{\rho} \frac{\partial p}{\partial x_i} + \frac{\mu}{\rho} \frac{\partial^2 u_i}{\partial x_j^2}, \tag{2}$$

where $u_i$ is the velocity components in the $i$ direction, $\rho$ is the fluid density, $\mu$ is the kinematic viscosity coefficient, and $p$ is the pressure in the flow field.

### 2.2. Numerical Model and Solution Setting

The computational domain is a rectangular region of $40L \times 20L$, where the side length of the square cylinder $L = 0.054$ m; the center of the square cylinder is located at the origin of the coordinates, as shown in Figure 1. The distances from the upstream inlet, the downstream outlet, and the two sides to the cylinder center are $10L$, $30L$, and $10L$, respectively. The selection of the above size parameters of the computational domain refers to the research results of Sohankar et al. [39] and Cheng et al. [40]. It meets the requirement that the blocking ratio is less than or equal to 5%, and the distance of $30L$ from model to outlet can ensure the full development of the downstream flow of the square cylinder. Two rotating circular cylinders with a diameter of $D$ are set on the rear corners of the square cylinder, and the distance between the rotating cylinders and the square cylinder is represented by $h$. The upper and lower sides of the square cylinder are tangential to the two circular cylinder surfaces.

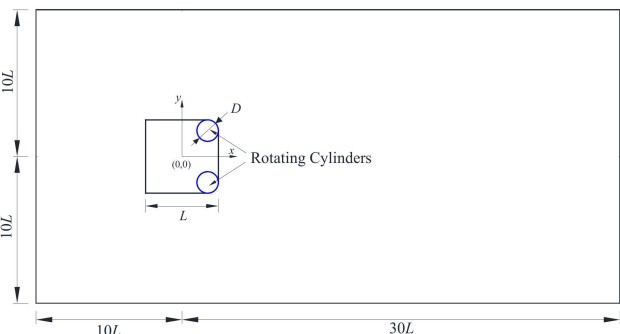

**Figure 1.** Schematic diagram of physical model and computational domain.

The computational domain is discretized using unstructured grids and is divided into two regions, namely Zone 1 and Zone 2, as shown in Figure 2. Zone 1 is the region around the two circular cylinders where the circular grids can rotate with the circular cylinders. The remaining region is Zone 2, and the fixed grids near the square cylinder surface are locally refined. The interface between the rotation grid (Zone 1) and fixed grid (Zone 2) is set as a slip boundary.

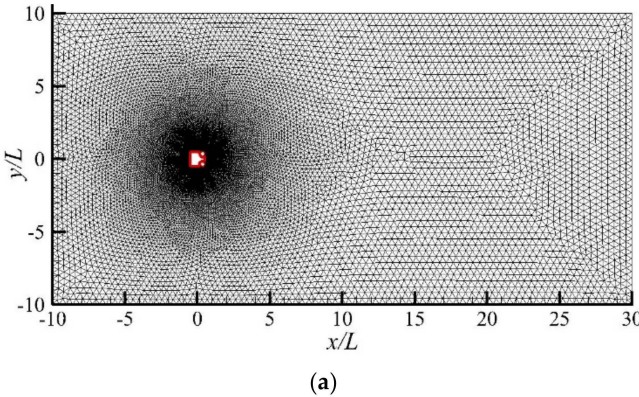
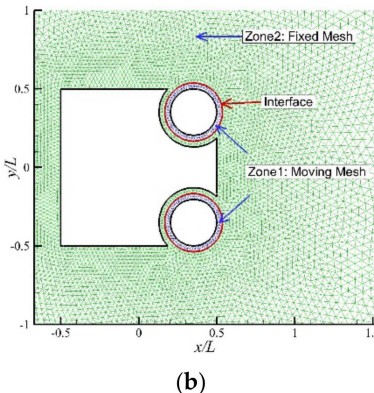

(**a**)
(**b**)

**Figure 2.** Computational mesh arrangement and domain partition, (**a**) Computational mesh, (**b**) Computational domain partition and local refined mesh.

The wind angle, rotation directions of the two circular cylinders, and arrangement of the pressure monitoring points for $h/D = 1/4$ and $D/L = 0.3$ are shown in Figure 3. The rotation directions of the two circular cylinders were designed as in the following three schemes: rotating in the downstream direction, rotating in the upstream direction, and co-rotating in the clockwise direction; each of the three schemes is represented in Figure 3a–c, respectively. The wind angle $\theta$ is defined as in the following: when the direction of the incoming airflow and the *x*-axis are the same, the wind angle is set to $0^\circ$, and the flow along the clockwise direction is considered as the positive direction. When the wind angle is $0^\circ$, the two rotating circular cylinders are placed at the two rear corners of the square cylinder. A total of 32 monitoring points were arranged on the model surface, out of which 18 points were evenly arranged on the straight edges of the square cylinder, and 7 points were evenly located on the quarter arc of each rotating circular cylinder. The main purpose is to obtain the pressure distribution characteristics on each straight edge of the square cylinder and on the two quarter arcs tangent to the surface of the square cylinder. In the calculation process, the pressure coefficient of each point was monitored to reveal the underlying mechanism of the MSBC method for manipulating the wake behind the square cylinder.

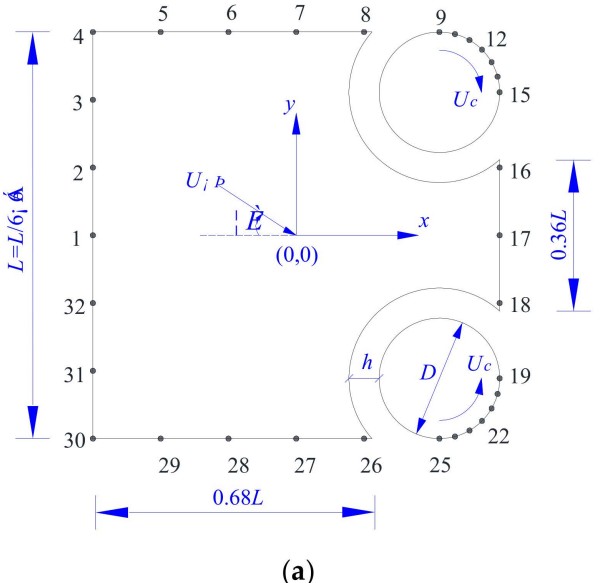
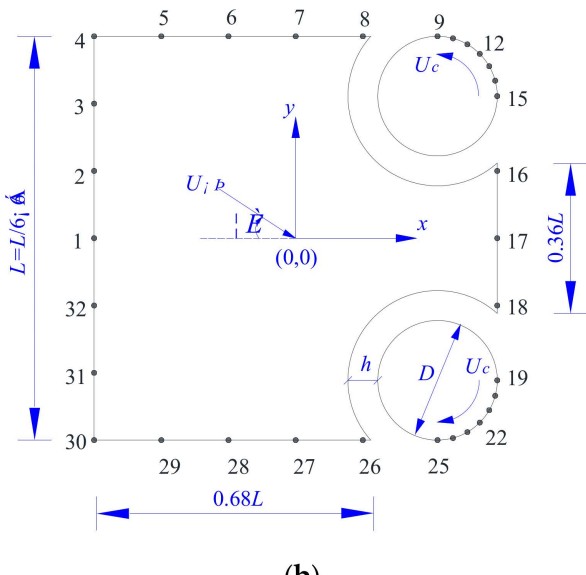

(**a**)
(**b**)

**Figure 3.** *Cont.*

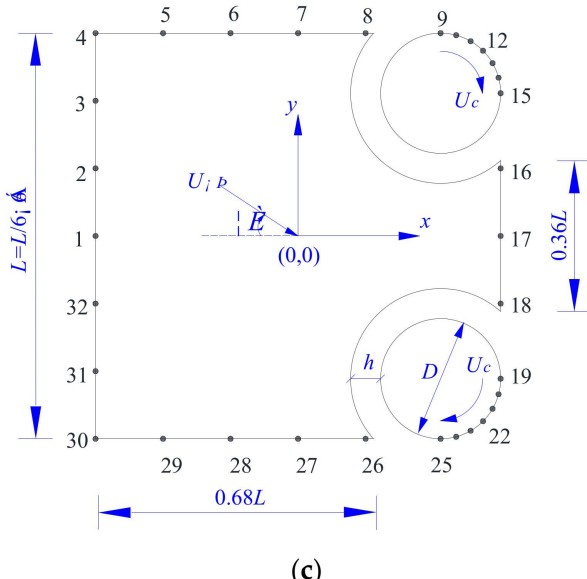

(**c**)

**Figure 3.** Wind angle, rotation direction and pressure monitoring point layout ($h/D = 1/4$, $D/L = 0.3$), (**a**) Inward rotation, (**b**) Outward rotation, (**c**) Clockwise co-rotation.

The velocity ratio $k$ ($k = U_c/U_\infty$) is the ratio of the tangential velocity of the rotating circular cylinder surface $U_c$ to the uniform incoming velocity $U_\infty$. The velocity ratio was adjusted by changing the rotation speed of the small cylinders, while the incoming airflow remained fixed. When $k = 0$, the two small circular cylinders were stationary; as the velocity ratio became higher, the momentum that the small cylinders delivered to the flow field became greater.

The flow direction was from the left to the right, and the following boundary conditions were set. The left side was a velocity inlet with a uniform velocity $U_\infty$ of 0.054 m/s; the right side was a pressure outlet, where the relative pressure was set to 0; and the upper and lower sides were set as symmetry boundaries. In addition, the square cylinder and the rotating circular cylinder surfaces were set as no-slip walls, and the interfaces between Zone 1 and Zone 2 were set as slip boundaries. The rotation of the circular cylinder and the surrounding mesh was realized through a dynamic mesh technique.

The numerical simulations were carried out at Re = 200, and the laminar model was employed. The finite volume method (FVM) was used to discrete the flow field around a square cylinder. The SIMPLE algorithm was used for the calculation of the coupling between the pressure and the velocity fields. The format of the pressure interpolation was selected to be 'Standard'. The second-order upwind scheme was utilized for the momentum discretization because of its stability and veracity. The time steps were changed according to the numerical simulation conditions. At the beginning, the dimensionless time step was set to 0.05. When the flow became fully developed and the vortex shedding occurred in the wake of the square cylinder, the MSBC became activated; then, the dimensionless time step was automatically reduced to 0.01.

### 2.3. Validity Investigation

The square cylinder with $\theta = 0°$ and $k = 0$ is defined as the 'standard square cylinder', whereas the square cylinder without rotating cylinder is referred to as a 'single square cylinder'. First, a mesh independence study was conducted in order to quantify the effect of mesh density on computational results. Simulations of flow around the single square cylinder were then performed at Re = 200 using four different mesh densities as listed in Table 1, where $N_c$, $N_{mesh}$, and $N_{nodes}$ represent, respectively, the node number on each side length of the square cylinder, the total number of grids, and nodes in the computational domain. The minimum grid size of each side of the single square cylinder $l_{min} = L/N_c$,

and the grids grow into the computational domain with each side of the square cylinder as source and the growth rate of 1.04, and the maximum grid size $l_{max} = 0.4L$. The comparison shown in Table 1 includes the fluctuation of lift coefficient ($C_l\prime$), the mean and fluctuation of drag coefficient ($\overline{C}_d$, $C_d\prime$), and the Strouhal number ($S_t$). As shown in Table 1, as the mesh density increases, the $C_d\prime$ and $S_t$ of different mesh schemes do not change significantly, and both $\overline{C}_d$ and $C_l\prime$ show a decreasing trend. The differences in highlighted aerodynamic coefficients $\overline{C}_d$ and $C_l\prime$ between the normal (Scheme 3) and dense (Scheme 4) meshes are 0.6 % and 4.2 %, and that between the coarse (Scheme 2) and dense (Scheme 4) meshes are 2.3 % and 12.8 %. Therefore, the normal mesh of 'Scheme 3' is appropriate in the following simulations of this study.

**Table 1.** Effect of mesh refinement on the calculation result of the flow around a single square cylinder at Re = 200.

| Mesh Density | $N_c$ | $N_{mesh}$ | $N_{nodes}$ | $\overline{C}_d$ | $C_d\prime$ | $C_l\prime$ | $S_t$ |
|---|---|---|---|---|---|---|---|
| Scheme 1, coarsest | 40 | 30,344 | 15,387 | 1.523 | 0.024 | 0.446 | 0.146 |
| Scheme 2, coarse | 55 | 36,208 | 18,349 | 1.489 | 0.022 | 0.404 | 0.148 |
| Scheme 3, normal | 70 | 42,092 | 21,323 | 1.465 | 0.020 | 0.373 | 0.148 |
| Scheme 4, dense | 85 | 48,590 | 24,602 | 1.456 | 0.019 | 0.358 | 0.149 |

Then, the mesh independent results of flow around the single square cylinder are compared with the existing research results to verify the accuracy of numerical simulation, and the results from the flow around the standard square cylinder are compared with those from the single square cylinder. The minimum grid size, growth ratio, and maximum grid size of each side of the standard square cylinder and the surfaces of the two rotating small cylinders are kept consistent with the parameters of the 'Scheme 3' meshing scheme of the single square cylinder. Figure 4 shows the time histories of the lift and drag coefficients of the single and standard square cylinders. The dimensionless lift coefficient by a spectral analysis is converted to the Strouhal number ($S_t$). The dimensionless time is $t^* = tU_\infty / L$. The alternating shedding vortex wake is the main cause of the periodic variation in the lift and drag coefficients, as shown in Figure 5. The figure shows the vorticity contours in the $z$ direction (perpendicular to the $x$-o-$y$ plane). The red color in the figure represents the positive vorticity of counterclockwise rotation, and the blue color represents the negative vorticity of clockwise rotation. The comparison between the results from the present numerical simulation and the experimental and numerical results from other previous work is presented in Table 2. The statistical results of aerodynamic parameters in the Table 2 are analyzed by using the time histories of lift and drag coefficients in the time period of $t^* = 80 \sim 150$, shown in Figure 4a. In the subsequent calculation cases, all pressure and aerodynamic coefficient time histories are calculated to the amplitude stable state, and the results of at least five vortex shedding characteristic periods are taken for statistical analysis. The simulation results of the aerodynamic forces and the Strouhal number are consistent with the previous results, and it indicates that the present grid accuracy, time step, and numerical settings can accurately capture the characteristics of the flow around a square cylinder. The mean drag coefficient $\overline{C}_d = 1.015$ of the standard square cylinder is 30.7% lower than that of the single square cylinder, the fluctuation drag coefficient $C_d\prime = 0.0128$ is 36.0% lower than that of the single square cylinder, and the fluctuation lift coefficient $C_l\prime = 0.324$ is 13.1% lower than that of the single square cylinder. The Strouhal number of the standard square cylinder ($S_t = 0.15$) is slightly higher than that of the single square cylinder. It is shown that a small change in the shape of the rear corner of the square cylinder can effectively reduce the aerodynamic force. However, the vortex shedding frequency will be slightly increased. In the following sections, the numerical results will be compared with the results of the flow around the standard square cylinder.

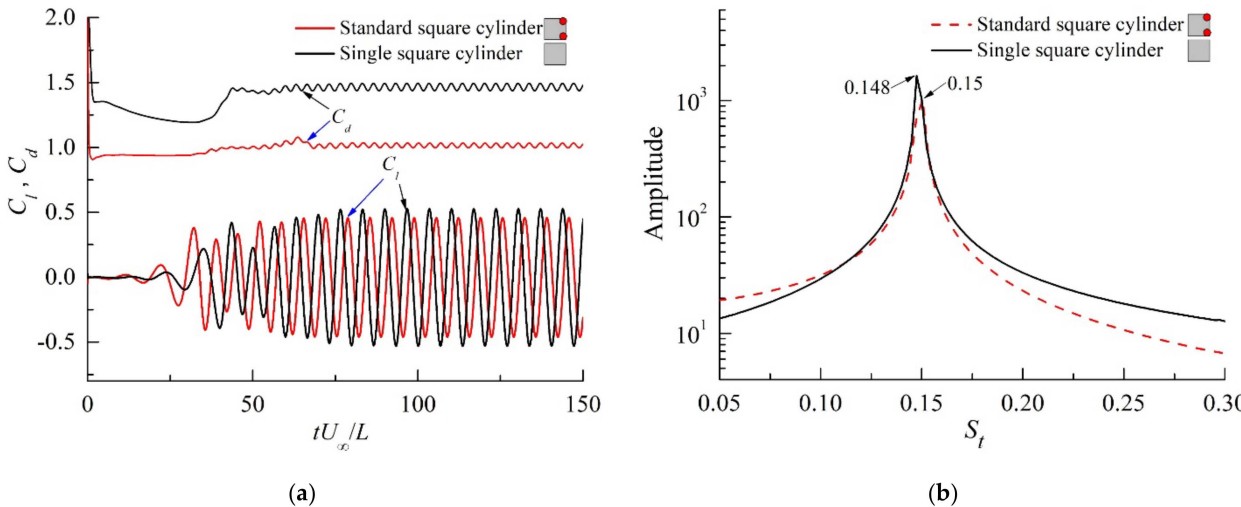

(**a**)                                  (**b**)

**Figure 4.** Comparisons of results of flow around a standard square cylinder and a single square cylinder, (**a**) Time histories of lift and drag coefficients, (**b**) Spectral analysis of $C_l$.

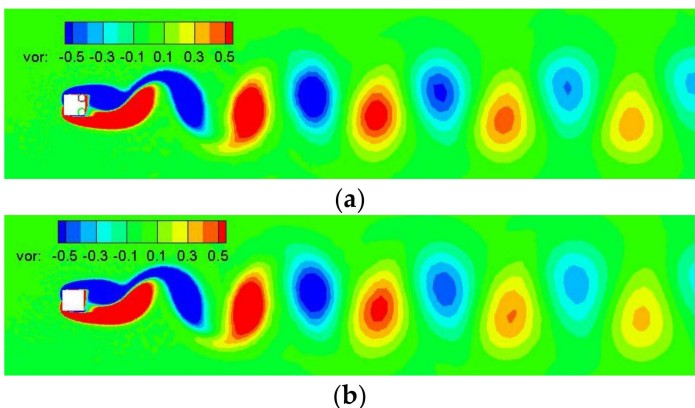

**Figure 5.** Vortex shedding in the wake of a standard square cylinder and a single square cylinder, (**a**) Standard square cylinder, (**b**) Single square cylinder.

**Table 2.** Comparison of the force coefficients statistics and Strouhal number of flow around a single square cylinder at Re = 200.

| Investigation | $\overline{C_d}$ | $C_d'$ | $C_l'$ | $S_t$ |
|---|---|---|---|---|
| Okajima [41], Experimental | 1.45 | - | - | 0.14~0.148 |
| Sohankar et al. [39], Numerical, 2D | 1.462 | - | 0.377 | 0.15 |
| Cheng et al. [40], Numerical, 2D | 1.45 | - | 0.372 | 0.15 |
| Jan and Sheu [42], Numerical, 2D | - | - | - | 0.148 |
| Abograis and Alshayji [43], Numerical, 2D | 1.488 | 0.027 | 0.332 | 0.153 |
| Present, Numerical, 2D ($N_c = 70$) | 1.465 | 0.020 | 0.373 | 0.148 |

## 3. Results and Discussion

### 3.1. Analysis of Influence of $D/L$ and $h/D$ Parameters

The diameter ($D$) of the small rotating circular cylinders and the spacing ($h$) between the rotating circular cylinders and the controlled square cylinder are important parameters that affect the control effect on the wake of the square cylinder. When wind angle $\theta = 0°$, velocity ratio $k = 2$, and the rotation direction of the circular cylinders is fixed to 'inward' rotation, as shown in Figure 3a, the influences of $D/L$ and $h/D$ on the statistical values and frequency characteristics of the aerodynamic coefficients of the square cylinder are mainly analyzed. The two dimensionless parameters $E_{C_l} = (C_l' - C_{l\_c}')/C_l'$ and

$E_{C_d} = (\overline{C}_d - \overline{C}_{d\_c})/\overline{C}_d$ are defined, which represent the relative variation in the RMS value of the lift coefficient and the mean value of the drag coefficient, respectively. These two parameters are used to evaluate the control effectiveness of the MSBC on the wake of flow around the square cylinder and determine the optimal parameters of the $D/L$ and $h/D$. When $E_{C_l}$ and $E_{C_d}$ are close to 1.0, the suppression effect on the wake is optimal. In contrast, when $E_{C_l}$ and $E_{C_d}$ are close to 0, no control exists.

Figure 6 shows the aerodynamic statistical parameters and Strouhal number ($S_t$ and $S_{t\_c}$) of flow around the uncontrolled ($k = 0$) and controlled ($k = 2$) square cylinder versus $D/L = 0.05 \sim 0.4$ when $h/D = 1/4$, together with the control effectiveness. The specific values corresponding to Figure 6 are shown in Table 3, where the aerodynamic statistical parameters include the RMS value of the lift coefficient ($C_l\prime$ and $C_{l\_c}\prime$), the mean value of the drag coefficient ($\overline{C}_d$ and $\overline{C}_{d\_c}$), and the RMS value of the drag coefficient ($C_d\prime$ and $C_{d\_c}\prime$). For different $D/L$ cases, the centers of the rotating circular cylinders can ensure that the side extended lines of the square cylinder are tangent to the circumference of the rotating circular cylinders. For the uncontrolled standard square cylinder, $C_l\prime$ is slowly decreased by 8%, $\overline{C}_d$ and $C_d\prime$ are decreased by 29% and 50%, respectively, while $S_t$ is kept unchanged at 0.15 with the increase of $D/L$ from 0.05 to 0.4. When there are two nonrotating cylinders at the corners of the square cylinder, it is equivalent to the rounded treatment of the leeward sharp corners of the square cylinder. This has less influence on the fluctuation degree of the lift coefficient in the cross-flow direction and vortex shedding frequency but can reduce mean value and fluctuation of the drag coefficient in the along-wind direction.

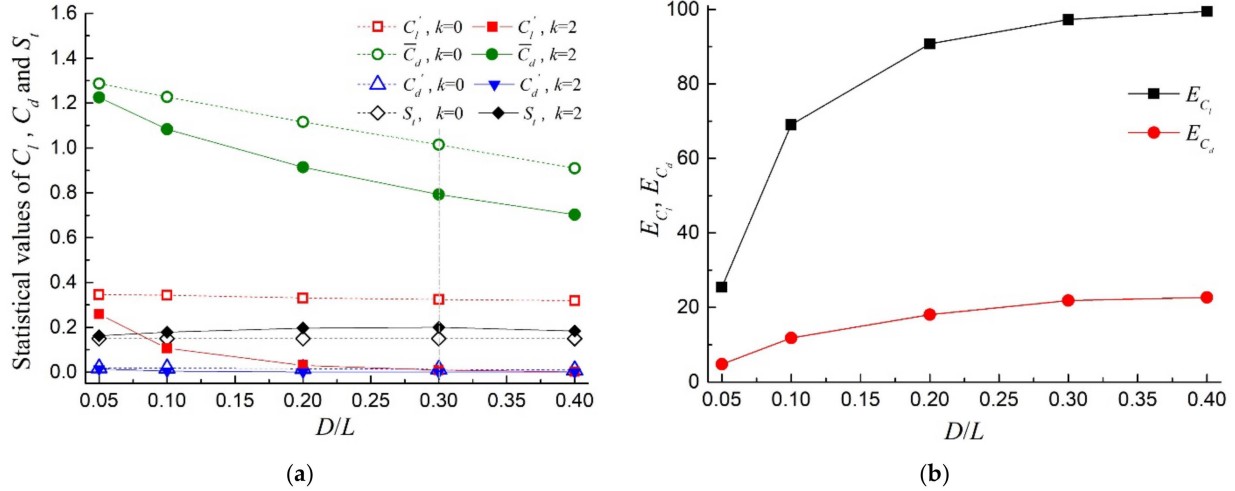

**Figure 6.** Statistical values of lift and drag coefficients, Strouhal number, and control effects under different $D/L$ values, (**a**) Statistical values of $C_l$, $C_d$, and $S_t$, (**b**) Control effects of $C_l$, $C_d$.

**Table 3.** Lift and drag coefficient statistics, Strouhal number, and control effects of flow around an uncontrolled ($k = 0$) and controlled ($k = 2$) square cylinder with $h/D = 1/4$.

| D/L | k=0 | | | | k=2 | | | | | |
|---|---|---|---|---|---|---|---|---|---|---|
| | $C_l\prime$ | $\overline{C}_d$ | $C_d\prime$ | $S_t$ | $C_{l\_c}\prime$ | $\overline{C}_{d\_c}$ | $C_{d\_c}\prime$ | $S_{t\_c}$ | $E_{C_l}$ (%) | $E_{C_d}$ (%) |
| 0.05 | 0.3466 | 1.2873 | 0.0186 | 0.15 | 0.2584 | 1.2255 | 0.0141 | 0.1625 | 25.45 | 4.80 |
| 0.1 | 0.3433 | 1.2274 | 0.0180 | 0.15 | 0.1063 | 1.0828 | 0.0051 | 0.1786 | 69.04 | 11.78 |
| 0.2 | 0.3309 | 1.1159 | 0.0155 | 0.15 | 0.0306 | 0.9144 | 0.0012 | 0.1961 | 90.75 | 18.06 |
| 0.3 | 0.3238 | 1.0145 | 0.0128 | 0.15 | 0.0089 | 0.7927 | 0.0008 | 0.2000 | 97.25 | 21.86 |
| 0.4 | 0.3189 | 0.9091 | 0.0093 | 0.15 | 0.0018 | 0.7027 | 0.0011 | 0.1832 | 99.44 | 22.70 |

When the two circular cylinders are rotating, the $C_{l\_c}\prime$, $\overline{C}_{d\_c}$, and $C_{d\_c}\prime$ of the controlled square cylinder are lower than those of the uncontrolled square cylinder, whereas the $S_{t\_c}$

of the controlled square cylinder is higher than that of the uncontrolled square cylinder. With the increasing $D/L$, the control effect $E_{C_l}$ on the RMS value of the lift coefficient is gradually increased from 25.45% to 99.44%, and the control effect $E_{C_d}$ on the mean value of the drag coefficient is gradually increased from 4.8% to 22.7%. When $D/L = 0.3$, the RMS value of drag coefficient plummets to its minimum value, which is 93.7% lower than that of the uncontrolled square cylinder, and the $E_{C_l}$ and $E_{C_d}$ at this time are 97.25% and 21.86%, respectively. When $D/L = 0.05 \sim 0.3$, the $S_{t\_c}$ is gradually increased from 0.1625 to 0.2; in turn, when the $D/L$ is increased to 0.4, the $S_{t\_c}$ is decreased slightly to 0.1832. It can be seen that the rotating circular cylinders play the role of controlling the wake of the square cylinder. The mean value of the drag coefficient can be reduced by up to 22.7%, and the fluctuations of the lift and drag coefficients are basically completely suppressed, while the vortex shedding frequency in the wake of the square cylinder is increased from 8.3% to 33.3%.

　　When $D/L = 0.3$, the aerodynamic statistical parameters, Strouhal number ($S_t$ and $S_{t\_c}$), and control effectiveness of flow around the uncontrolled ($k = 0$) and controlled ($k = 2$) square cylinder versus $h/D$, the range of which varies from 1/8 to 3/8 with an interval of 1/16, are shown in the Figure 7 and Table 4. With gradually increasing $h/D$, the $C_l\prime$ of the uncontrolled square cylinder does not change significantly between 0.3226 and 0.3388, and the $S_t$ is also kept unchanged at 0.15, while $\overline{C}_d$ and $C_d\prime$ are decreased by 13% and 23%, respectively. When $k = 2$, the $C_{l\_c}\prime$ and $\overline{C}_{d\_c}$ of the controlled square cylinder are gradually increased with increase of $h/D$, and the corresponding control effects of $E_{C_l}$ and $E_{C_d}$ are gradually decreased with the increasing $h/D$. The $C_{d\_c}\prime$ of controlled square cylinder changes from 0.0006 to 0.0039, which is about 5.2% to 26% of the results of uncontrolled square cylinder, and the $S_{t\_c}$ of controlled square cylinder is between 0.18 and 0.2, which is still greater than the result of the uncontrolled square cylinder.

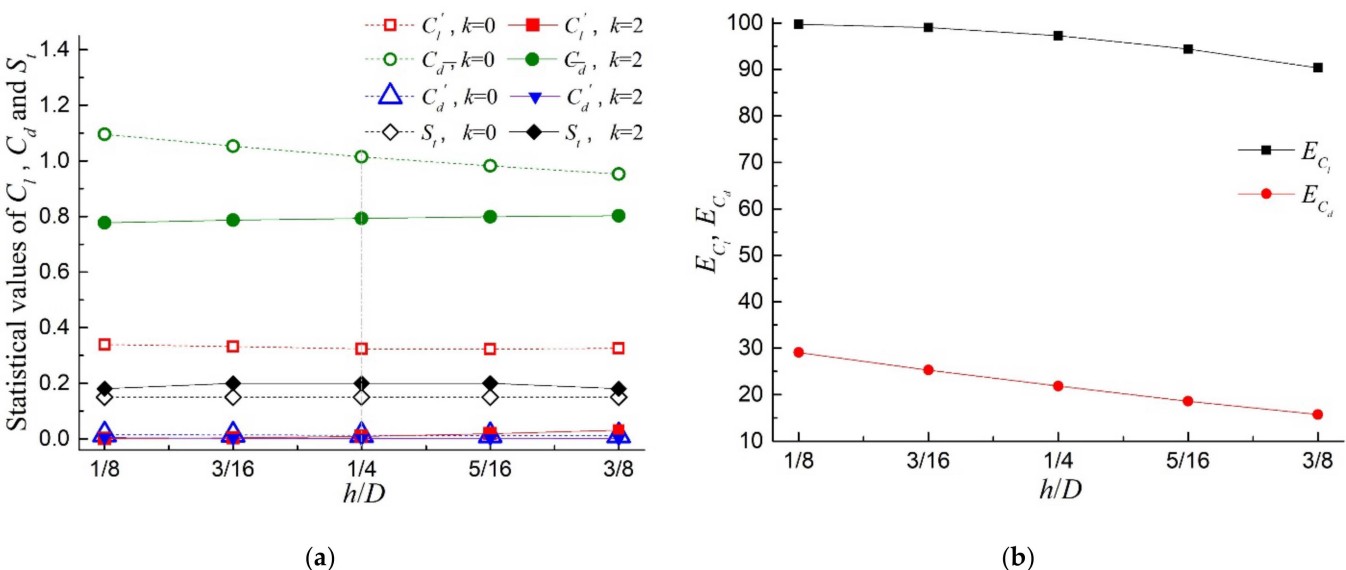

(**a**)　　　　　　　　　　　　　　　　　　　　　　　　(**b**)

**Figure 7.** Statistical values of lift and drag coefficients, Strouhal number, and control effects under different $h/D$ values, (**a**) Statistical values of $C_l$, $C_d$, and $S_t$, (**b**) Control effects of $C_l$, $C_d$.

**Table 4.** Lift and drag coefficient statistics, Strouhal number, and control effects of flow around an uncontrolled ($k = 0$) and controlled ($k = 2$) square cylinder with $D/L = 0.3$.

| D/L | k=0 | | | | k=2 | | | | | |
|---|---|---|---|---|---|---|---|---|---|---|
| | $C'_l$ | $\overline{C}_d$ | $C'_d$ | $S_t$ | $C'_{l\_c}$ | $\overline{C}_{d\_c}$ | $C'_{d\_c}$ | $S_{t\_c}$ | $E_{C_l}$ (%) | $E_{C_d}$ (%) |
| 1/8 | 0.3388 | 1.0956 | 0.0150 | 0.15 | 0.0009 | 0.7770 | 0.0039 | 0.18 | 99.73 | 29.08 |
| 3/16 | 0.3318 | 1.0534 | 0.0140 | 0.15 | 0.0032 | 0.7870 | 0.0020 | 0.20 | 99.04 | 25.29 |
| 1/4 | 0.3238 | 1.0145 | 0.0128 | 0.15 | 0.0089 | 0.7927 | 0.0008 | 0.20 | 97.25 | 21.86 |
| 5/16 | 0.3226 | 0.9821 | 0.0121 | 0.15 | 0.0181 | 0.7996 | 0.0013 | 0.20 | 94.39 | 18.58 |
| 3/8 | 0.3254 | 0.9524 | 0.0115 | 0.15 | 0.0314 | 0.8026 | 0.0006 | 0.18 | 90.35 | 15.73 |

Through comparative analysis, we can see that the smaller the spacing (*h*) between the rotating circular cylinders and the square cylinder, the better the control effect of the MSBC method on the wake of the square cylinder. As the spacing ratio $h/D$ decreases from 3/8 to 1/8, the results of $E_{C_l}$ and $E_{C_d}$ are increased by 9.38% and 13.35%, respectively, which are less than the increasing extents of $E_{C_l}$ and $E_{C_d}$ with increase of $D/L$ and are especially far less than the increasing extent of 73.99% for $E_{C_l}$, as shown in Table 2. It can be seen that the influence of the diameter of the rotating circular cylinders on the control effect of the wake of the square cylinder is greater than the influence of the variation of the spacing between the rotating circular cylinders and the square cylinder on the control effect.

Compared with the control effects of $D/L = 0.3$ and $h/D = 1/4$, when $D/L$ continues to increase from 0.3 to 0.4, the improvements of the control effects $E_{C_l}$ and $E_{C_d}$ of the controlled square cylinder are not obvious, only increased by 2.19% and 0.84%. As the $h/D$ is decreased from 1/4 to 1/8 continually, the $E_{C_l}$ is only increased from 97.25% to 99.73%, and the $E_{C_d}$ is only increased by 7.22% at the same time. On the contrary, the fluctuation of the drag coefficient $C_{d\_c}\prime$ is increased from 6.3% to its maximum of 26% of $C_d\prime$ for the uncontrolled standard square cylinder. From the above analysis, it can be concluded that $D/L = 0.3$ and $h/D = 1/4$ are the better combination of the parameters to achieve a better control effect on the wake of flow around the square cylinder.

*3.2. Detailed Results and Analysis of $D/L = 0.3$ and $h/D = 1/4$*

The present numerical simulation starts from the flow around a fixed square cylinder; when the alternating vortex shedding street has been formed behind the square cylinder, the MSBC becomes activated. The rotation directions of the circular cylinders include the following three forms: inward rotation, outward rotation, and clockwise co-rotating, as shown in Figure 3. The wind angle range was $0° \sim 180°$ and the angle interval was $30°$. For the inward rotation case, an extra wind angle of $15°$ was added. The velocity ratio range was $0 \sim 4.0$. The aerodynamic coefficients of the square cylinder under different wind angles and velocity ratios were calculated. Then, the statistical and frequency characteristics of the aerodynamic coefficients were obtained. In the following sections, the pressure coefficient and the vortex shedding mode are also presented.

3.2.1. Aerodynamic Statistics and Frequency Characteristics

Figures 8–10 show the variations in the lift and drag coefficient statistics of the square cylinder for the wind angle ($\theta$) and the velocity ratio ($k$) for each of the three rotation directions, respectively. The results include the mean value $\overline{C}_l$ and the fluctuating value $C_l\prime$ of the lift coefficient, as well as the mean value $\overline{C}_d$ and the fluctuating value $C_d\prime$ of the drag coefficient. When $k > 0$, the small rotating cylinders transfer the momentum into the flow field. The results for $k = 0$ given in the abovementioned figures correspond to the results from the flow around a standard square cylinder. As shown in Figure 8a, when $\theta = 0°$ and $\theta = 180°$, the square cylinder is symmetrically placed in the flow direction. Therefore, the value of $\overline{C}_l$ is near zero for different $k$ values at these two wind angles. When $15° < \theta < 90°$ and $k > 0$, the absolute value of $\overline{C}_l$ is greater than the absolute value of $\overline{C}_l$ when $k = 0$. When $\theta = 15°$ and $k = 4$, the value of $\overline{C}_l$ increases 5.36 times compared with its value when

$k = 0$. When $\theta \geq 120^{\circ}$, the values of $\overline{C}_l$ are similar among them for various $k$ values. As illustrated in Figure 8b, when $\theta \leq 60^{\circ}$ and $k > 0$, $C_l\prime$ is lower than the value of $C_l\prime$ when $k = 0$. It may be observed that for smaller wind angles, the rate of decrease of $C_l\prime$ becomes more obvious. The alternate shedding vortex behind the square cylinder is responsible for the lift coefficient fluctuation; the results show that the vortex shedding of the square cylinder should be suppressed at small wind angles. When $k = 4$ and $\theta$ increases from $0^{\circ}$ to $15^{\circ}$, $C_l\prime$ decreases by 98% and 87%, respectively, compared with the value of $C_l\prime$ when $k = 0$. When $\theta = 60^{\circ}$ and $k = 4$, the value of $C_l\prime$ is slightly higher than the value of $C_l\prime$ when $k = 2$. This instance suggests that the increment of momentum that is transferred to the flow field may lead to the diminution of the control effect. When $\theta > 60^{\circ}$, $C_l\prime$ increases with the increase in $k$; this means that the MSBC is involved in the enhancement of the wake. As shown in Figure 8c, the variation of $\overline{C}_d$ at $k = 0$ is similar to that of $C_l\prime$ at $k = 0$ in Figure 8b. When $k = 4$ and $\theta$ increases from $0^{\circ}$ to $15^{\circ}$ and then increases to $30^{\circ}$, the values of $\overline{C}_d$ decrease by 36%, 54%, and 47%, respectively, compared with the value of $\overline{C}_d$ when $k = 0$. When $\theta = 60^{\circ}$, the value of $\overline{C}_d$ at $k = 2$ decreases to the lowest point. When $\theta = 90^{\circ}$ and $\theta = 120^{\circ}$, the values of $\overline{C}_d$ at $k = 4$ are 1.94 and 1.38 times the values of $\overline{C}_d$ when $k = 0$, respectively. When $\theta = 150^{\circ}$ and $k > 0$, the values of $\overline{C}_d$ are near the value that corresponds to $k = 0$. When $\theta = 180°$, the values of $\overline{C}_d$ at $k = 2$, 4 increase by 37% and 44%, respectively. As shown in Figure 8d, when $\theta \geq 30^{\circ}$, $C_d\prime$ increases with the increase in $k$, and the change is similar to the results when $k = 0$. When $\theta = 15^{\circ}$, the values of $C_d\prime$ for $k > 0$ can decrease by 25% compared to the value of $C_d\prime$ for $k = 0$.

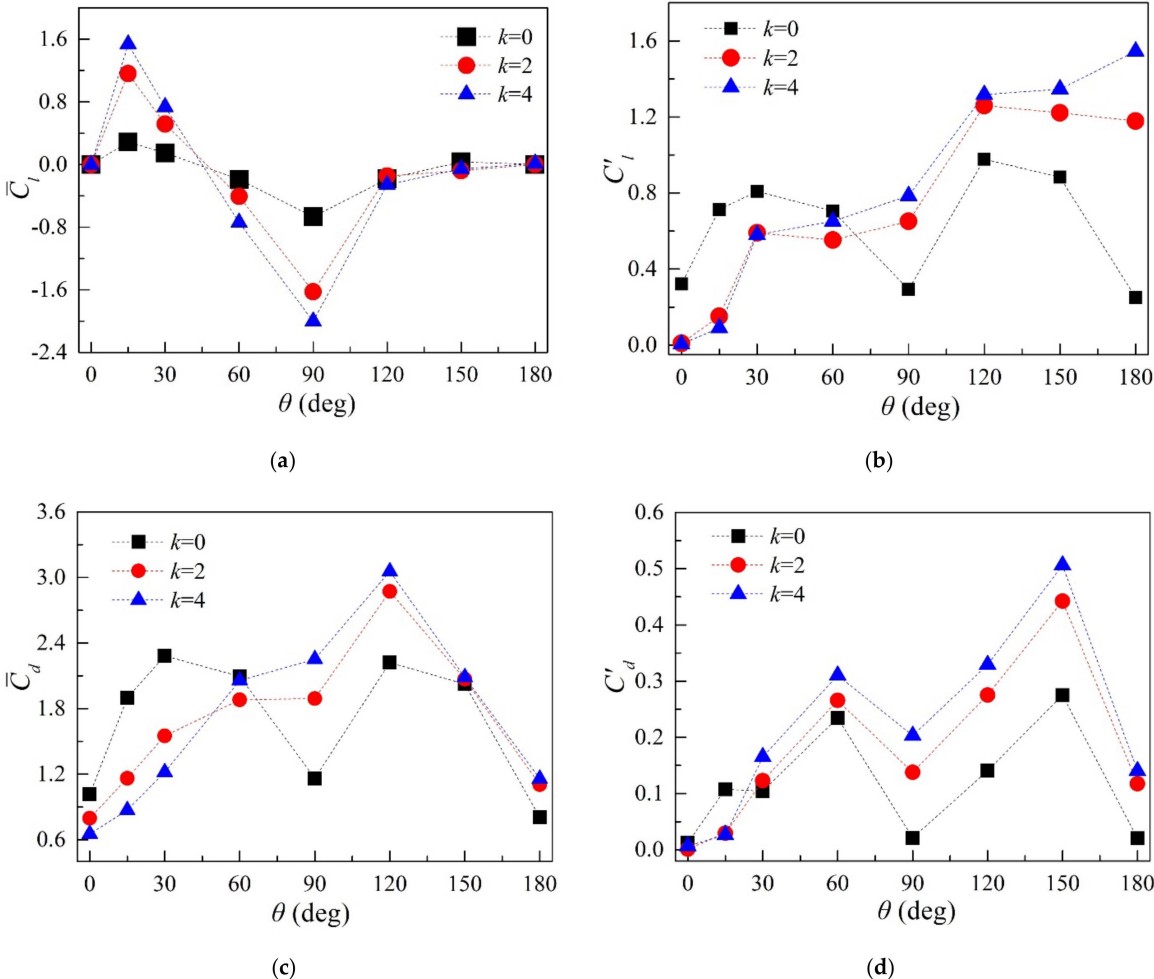

**Figure 8.** Aerodynamic coefficient statistics of the square cylinder for the wind angles ($\theta$) and the velocity ratios ($k$) under inward rotation, (**a**) $\overline{C}_l$, (**b**) $C_l\prime$, (**c**) $\overline{C}_d$, (**d**) $C_d\prime$.

Through the above analysis, it may be observed that the control method may help to improve the wake stability of a square cylinder when $\theta \leq 60°$, and it is unfavorable to the stability of the wake for the remaining wind angles. The control effect on the oscillating wake of the square cylinder is optimal when $\theta = 0°$. When $\theta = 15°$, this particular control method resulted in a great decrease in $\overline{C}_d$ and a great increase in $\overline{C}_l$ of the square cylinder. From this viewpoint, the optimum effect of 'drag reduction and lift increment' is achieved at this particular wind angle.

In Figure 9, the variation of the statistical parameters of the aerodynamic forces with wind angles is similar for $k = 2$ and $k = 4$; the difference lies only in terms of the individual angle. When $k = 0$, the aerodynamic parameters of the square cylinder (outward rotation) are consistent with the results in Figure 8. When $\theta = 0°$, the values of $\overline{C}_l$ for different $k$ values are near 0; however, the value of $C_l\prime$ increases from 0.32 at $k = 0$ to 0.97 at $k = 4$. In a similar manner, the mean and fluctuating values of the drag coefficient have increased significantly, thus suggesting that the momentum generated by the rotating cylinder leads to a more intense vortex shedding in the wake of the square cylinder. When $\theta = 30°$ and $\theta = 60°$, $\overline{C}_l$ is not equal to 0, and $C_l\prime$, $\overline{C}_d$, and $C_d\prime$ present growth trends as the $k$ value incrementally increases; this illustrates that the wake vortex pattern has changed, and the MSBC presents an effect of 'lift and drag increment'. When $\theta = 90°$ and $k = 4$, $C_l\prime$ and $C_d\prime$ increase slightly compared with their values when $k = 0$. $\overline{C}_d$ remains unchanged, and $\overline{C}_l$ slightly decreases. This indicates that the momentum injection does not improve the flow structure. When $\theta = 120°$ and $\theta = 150°$, $\overline{C}_l$ increases significantly with the increase in $k$; however, $C_l\prime$, $\overline{C}_d$, and $C_d\prime$ present a significantly decreasing trend. This suggests that the oscillating wake of the square cylinder has been suppressed and that the MSBC presents an effect of 'drag reduction and lift increment'. When $\theta = 120°$ and $k = 4$, $\overline{C}_d$ decreases by 90% compared with the value of $\overline{C}_d$ when $k = 0$, and $C_d\prime$ decreases by 60%; therefore, the 'drag reduction' effect is obvious. In addition, the value of $\overline{C}_l$ becomes six times higher than the value of $\overline{C}_l$ when $k = 0$, whereas $C_l\prime$ decreases by 75%; thus, the 'lift increment' effect is obvious. When $\theta = 180°$, $\overline{C}_d$ increases slightly with the increase in $k$; the values of the remaining parameters do not present significant differences with their respective values when $k = 0$. The results show that the MSBC method has no obvious effect on the oscillating wake for this particular wind direction.

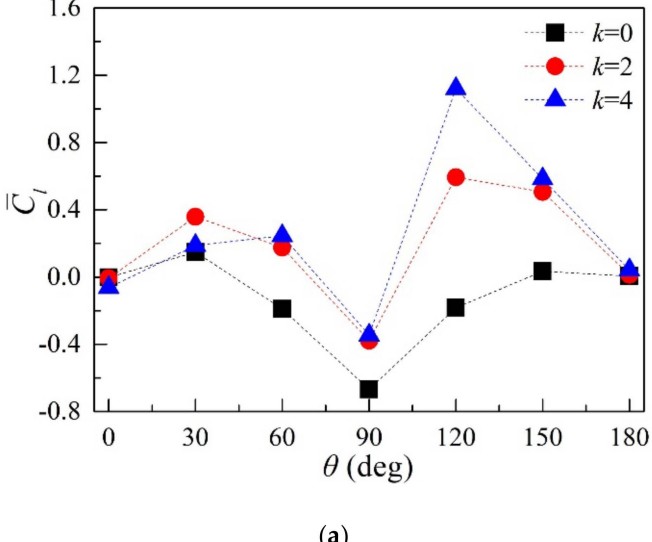

(**a**)

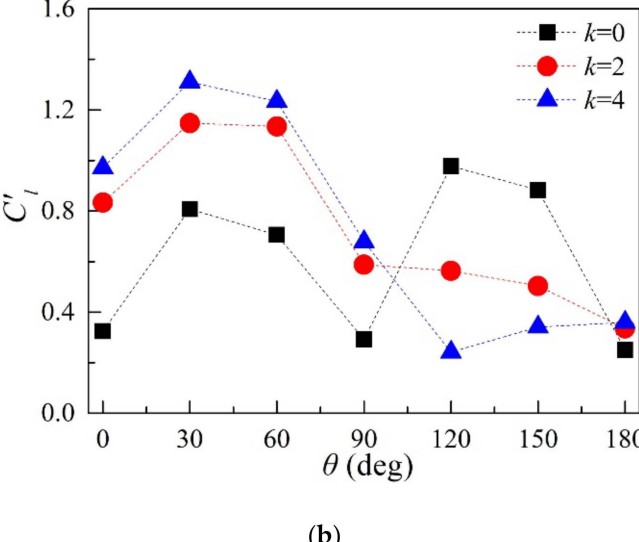

(**b**)

**Figure 9.** *Cont.*

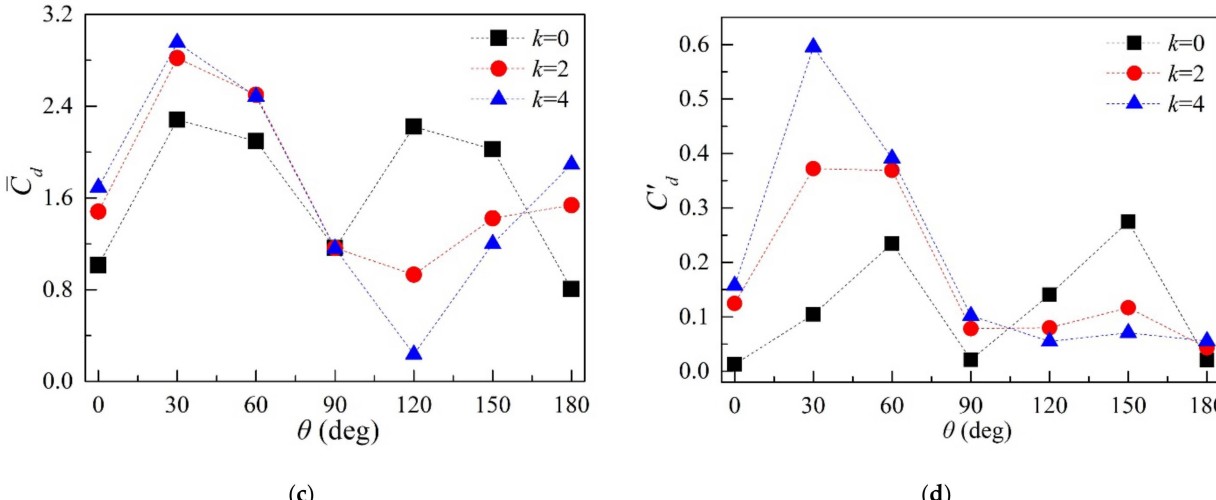

**Figure 9.** Aerodynamic coefficient statistics of the square cylinder for the wind angles (θ) and the velocity ratios (*k*) under outward rotation, (**a**) $\overline{C}_l$, (**b**) $C_l\prime$, (**c**) $\overline{C}_d$, (**d**) $C_d\prime$.

As shown in Figure 10, when $k > 0$, the aerodynamic statistics of the square cylinder present a similar change for different wind angles. With the increase in *k* from 0 to 4, the momentum of the rotation that the circular cylinders transfer to the flow field gradually increases. When $\theta = 0°$, $\overline{C}_l$ increases from 0 to 1.6, and $C_l\prime$ increases approximately by 100%; $\overline{C}_d$ remains almost unchanged, whereas $C_d\prime$ slightly increases. When $\theta = 30°$ and $\theta = 60°$, $\overline{C}_l$ at $k = 4$ increases to 2.1 and 1.1, respectively, whereas $C_l\prime$ decreases by 36% and 90%, respectively. This reveals that the oscillating wake of the square cylinder can be suppressed to different levels for these particular two wind angles. Although the fluctuation of the lift coefficient has been eliminated, the mean of the lift coefficient has increased. Simultaneously, the value of $\overline{C}_d$ decreased by 54% and 58% for $\theta = 30°$ and $\theta = 60°$, respectively. The values of $C_d\prime$ at $\theta = 30°$ slightly decrease with the increase in *k*, whereas $C_d\prime$ at $\theta = 60°$ decreases by 87%. When $\theta = 90°$, $C_l\prime$ at $k = 2$ and $k = 4$ decreases to 0; this reveals that the alternate shedding vortices in the wake of the square cylinder have been eliminated. When $\theta = 120°$ and $\theta = 150°$, $\overline{C}_l$ still presents a growth trend, whereas the remaining statistical parameters present lower values than their corresponding values when $k = 0$; therefore, the input of momentum presents an effect of 'drag reduction and lift increment'. When $\theta = 180°$, $\overline{C}_l$ presents a negative increment, whereas the remaining statistics present higher values than their corresponding values when $k = 0$. This demonstrates that the input of momentum causes the wake vortex shedding to become enhanced in this particular wind direction.

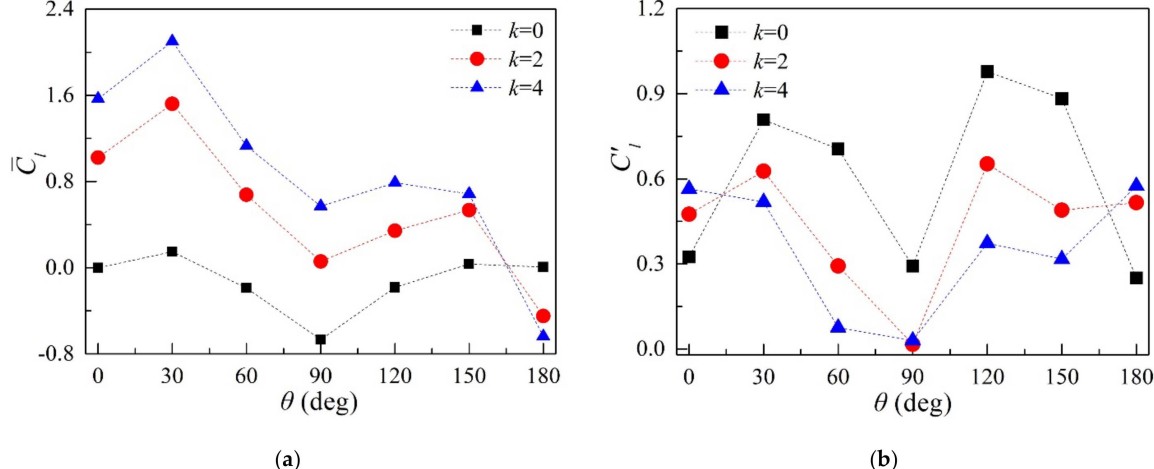

**Figure 10.** *Cont.*

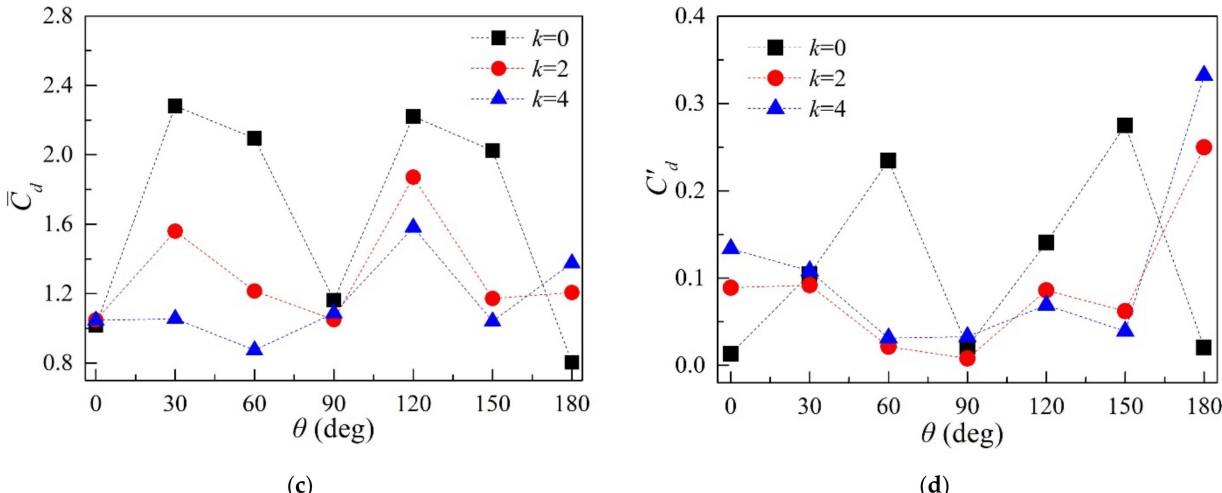

**Figure 10.** Aerodynamic coefficient statistics of the square cylinder for the wind angles ($\theta$) and the velocity ratios ($k$) under co-rotation, (**a**) $\overline{C}_l$, (**b**) $C_l\prime$, (**c**) $\overline{C}_d$, (**d**) $C_d\prime$.

Figures 11–13 show the dimensionless amplitude spectra of the lift coefficients of the square cylinder for different velocity ratios and different wind angles for the cases of inward rotation, outward rotation, and clockwise co-rotation, respectively. The two horizontal axes represent the wind angle and the Strouhal number, respectively; the vertical axis represents the amplitude of the fast Fourier transform (FFT). The control effect of the MSBC method on the oscillating wake of the square cylinder can be observed through the maximum amplitude for different $k$ values; simultaneously, the high harmonics with small amplitudes caused by the input of momentum can be observed, as well. When the circular cylinders start to rotate, the amplitude of certain spectral analysis curves decreases, or the peak disappears completely. This shows that the alternate shedding vortex street of the square cylinder wake is suppressed or eliminated under this wind direction.

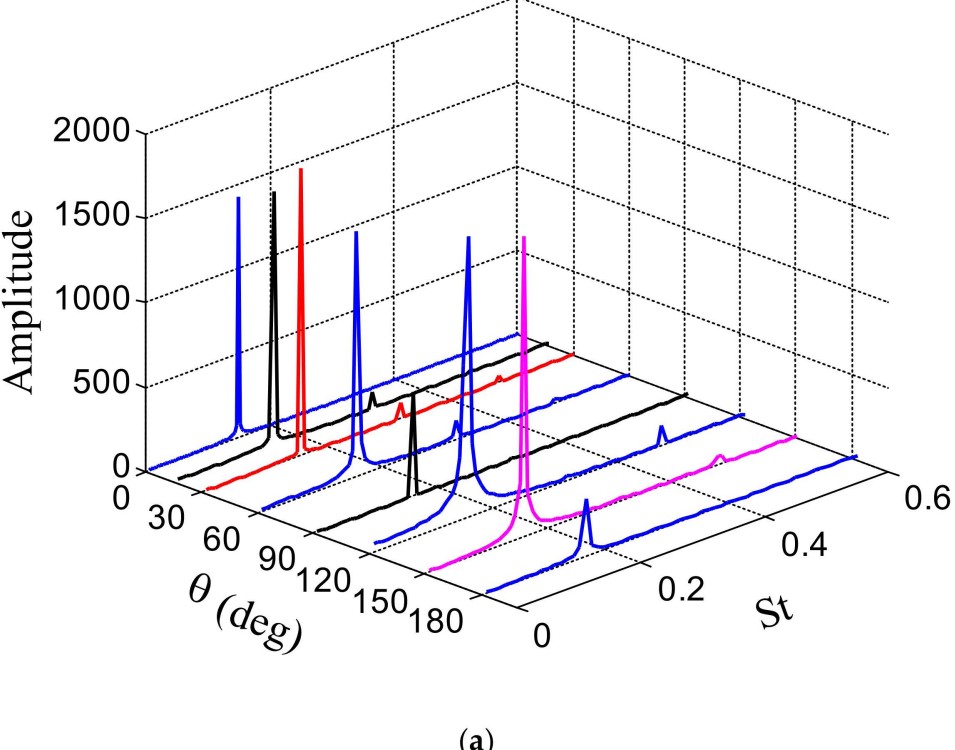

(**a**)

**Figure 11.** *Cont.*

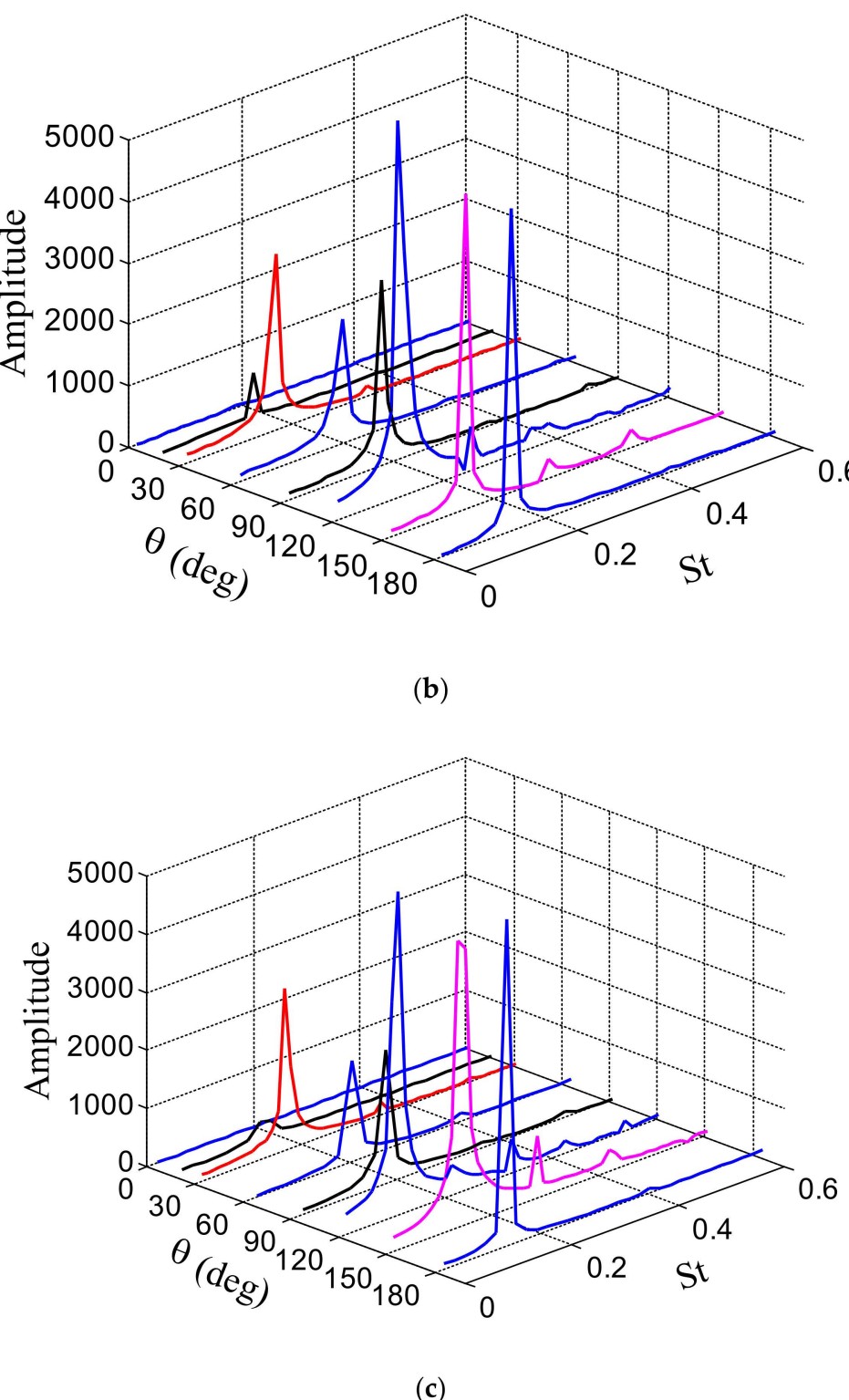

**Figure 11.** Amplitude spectra of Strouhal number under different wind angles with the change of velocity ratios (inward rotation), (**a**) $k = 0$, (**b**) $k = 2$, (**c**) $k = 4$.

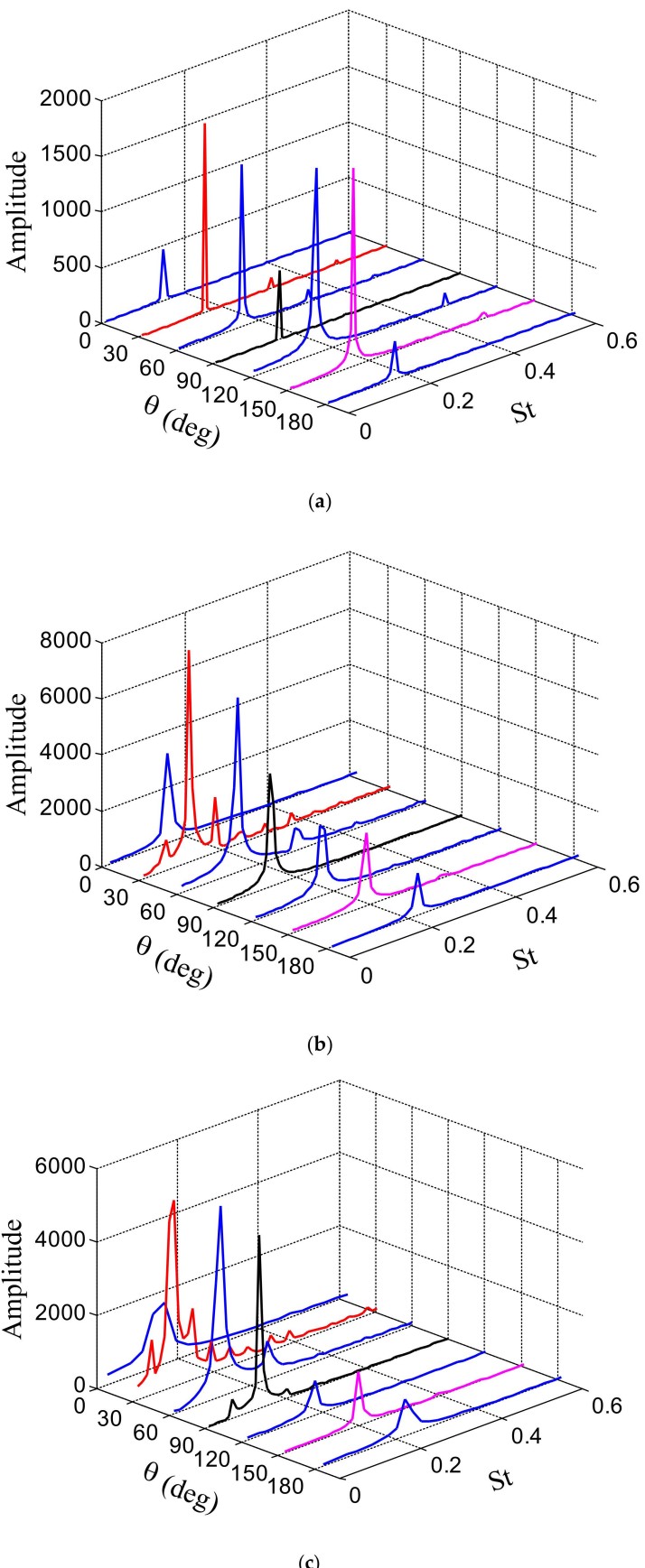

**Figure 12.** Amplitude spectra of Strouhal number under different wind angles with the change of velocity ratios (outward rotation), (**a**) $k = 0$, (**b**) $k = 2$, (**c**) $k = 4$.

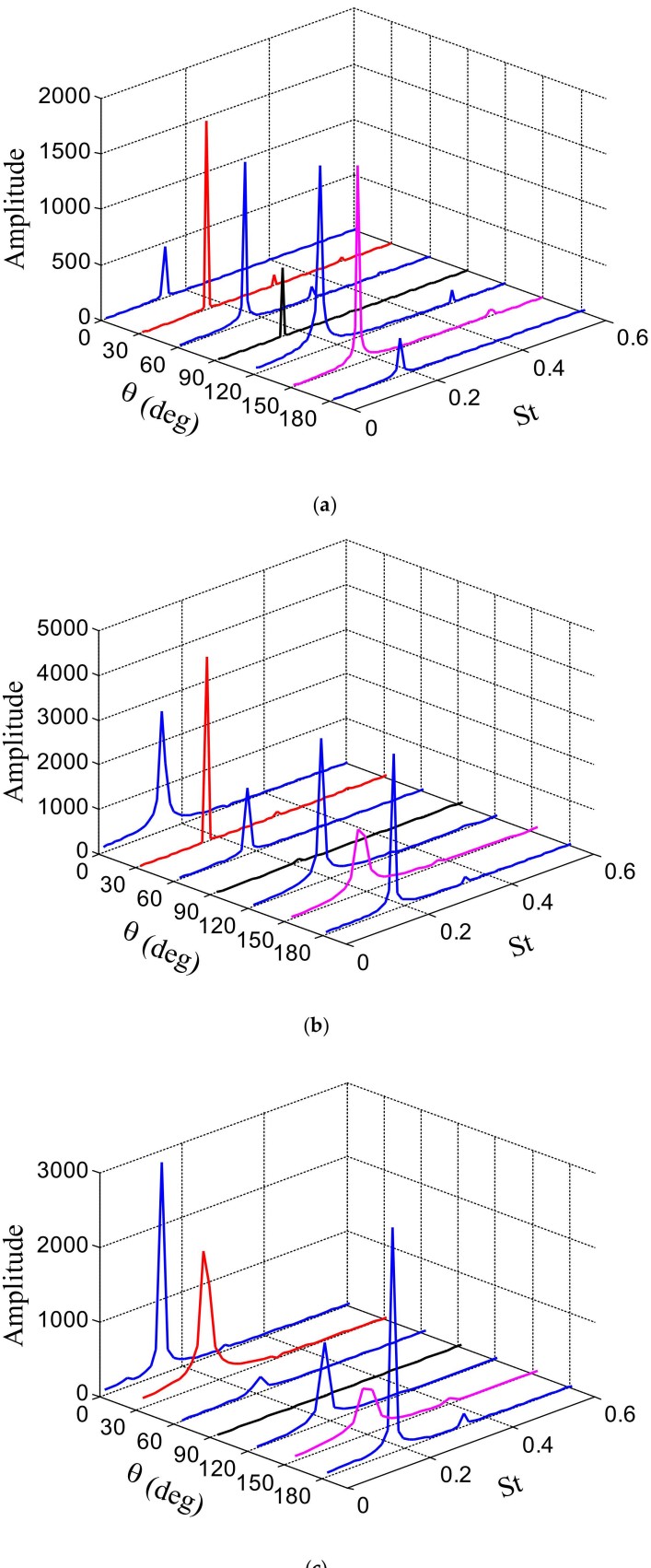

(**a**)

(**b**)

(**c**)

**Figure 13.** Amplitude spectra of Strouhal number under different wind angles with the change of velocity ratios (co-rotation), (**a**) $k = 0$, (**b**) $k = 2$, (**c**) $k = 4$.

The corresponding Strouhal numbers of the peak values of each curve in Figures 11–13 were calculated and are shown in Figure 14. As can be seen from Figure 14a, the values of the vortex shedding frequency of the square cylinder are close to each other for $k = 2$ and $k = 4$, when the corner circular cylinders rotate inward. When $\theta = 0°$, $S_t$ is near 0; the main reason for this is that the alternate shedding vortex has been completely eliminated. When $\theta = 30° \sim 90°$, the vortex shedding frequency is greater than that at $k = 0$, whereas when $\theta = 120° \sim 180°$, the vortex shedding frequency is lower than that at $k = 0$. When the circular cylinders rotate outward, the MSBC method cannot eliminate the wake vortex street behind the square cylinder in the entire range of the wind angles. As shown in Figure 14b, when $\theta = 30° \sim 90°$, the vortex shedding frequency is lower than that at $k = 0$. For the remaining wind angles, the values of the vortex shedding frequency are higher than their corresponding values at $k = 0$. As illustrated in Figure 14c, when $\theta = 60°$ and $k = 4$, $S_t$ is equal to 0; this means that the wake vortex street behind the square cylinder has been entirely eliminated. In addition, when $\theta = 90°$, $k = 2$, and $k = 4$, the wake vortex street is eliminated, as well, and the MSBC method presents an improved flow control effect.

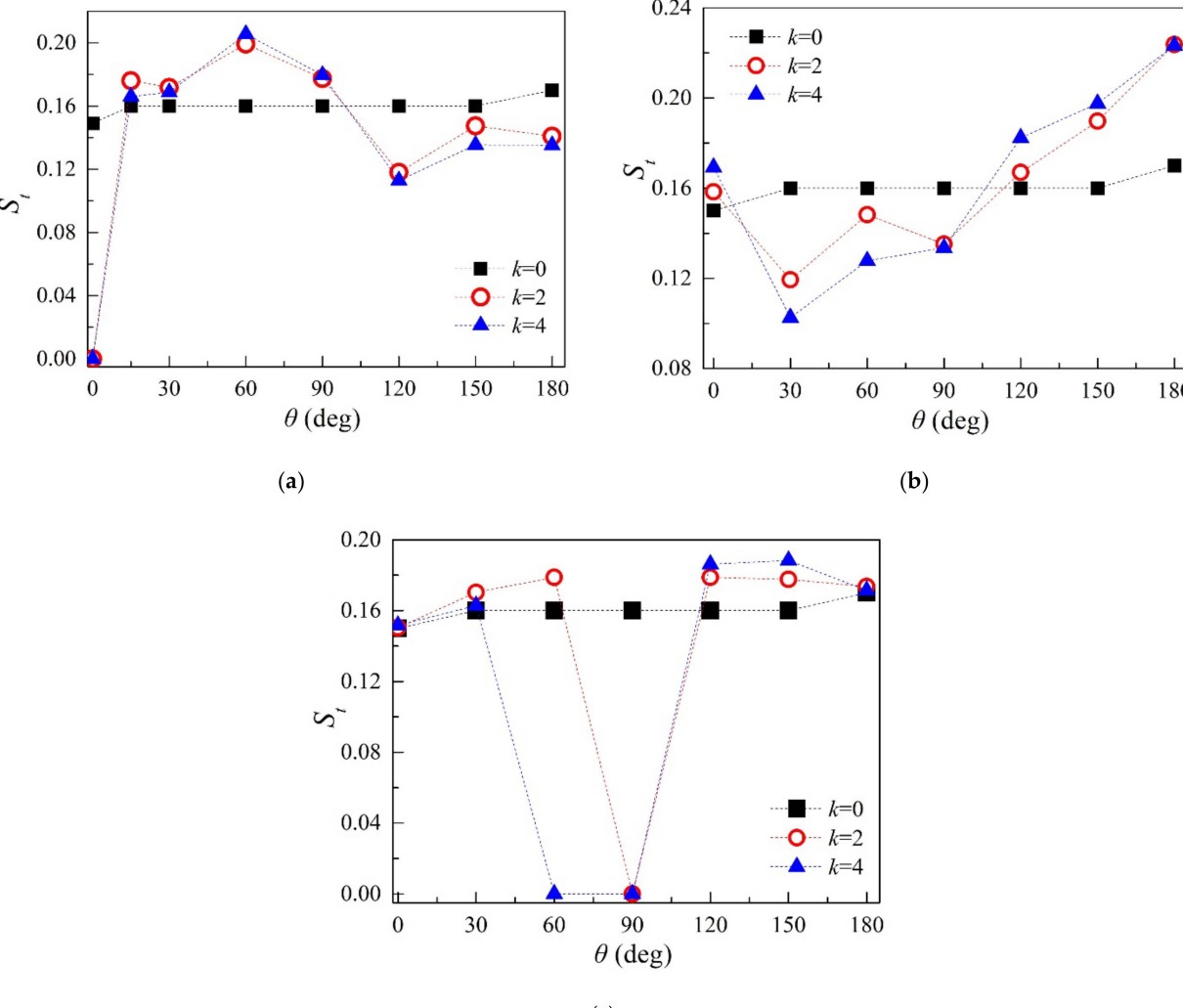

**Figure 14.** Variation of Strouhal number for the velocity ratios and the wind angles under different rotation modes, (**a**) Inward rotation, (**b**) Outward rotation, (**c**) Clockwise co-rotation.

### 3.2.2. The Mean Pressure Distribution Characteristics

Figures 15–17 show the distribution characteristics of the mean pressure coefficients around the square cylinder for different rotation modes, different $k$ values, and wind angles

$\theta$. Figure 15 shows the distribution of the mean pressure coefficient on the surface of the square cylinder at each wind angle, when the cylinder rotates inward. According to the analysis results in Figure 8, the oscillating wake of the square cylinder is suppressed to a different level when $\theta < 30^{\circ}$. Therefore, the corresponding mean pressure distributions on the surface of the square cylinder at $k = 1 \sim 4$ are illustrated in detail in Figure 15a–c. For the remaining wind angles, only the results for $k = 2$ and $k = 4$ are presented, as shown in Figure 15d–h. When $\theta = 0^{\circ}$, the mean pressure coefficient around the square cylinder is symmetrically distributed. At the windward side of the square cylinder is the flow stagnation zone, resulting in a positive pressure zone on the windward side, and the positive wind pressure reaches its maximum value at point No. 1. The flow separates from the leading edge corner of the square cylinder into the wake region and develops to an alternating shedding vortex street, thus leading to higher negative pressures at points No. 4 and 30. The measuring points No. 9–15 and 19–25 are located around the rotating circular cylinders, and points No. 16–18 are located on the surface of the leeward side between the two circular cylinders. Compared with the mean pressures when $k = 0$, when the small cylinder begins to rotate, the mean pressures of the measuring points No. 9–25 have changed significantly; however, there are small differences among the mean pressures at the remaining measuring points. This suggests that the circular cylinder rotation has a small effect on the upstream flow of the square cylinder. The negative pressure coefficients on the two small circular cylinders increase with the incremental increase in $k$. Owing to the suction generated by the negative pressure, the shear layers that have separated from the leading edge attach to the upper and lower sides of the square cylinder, and steadily flow into the downstream without vortex formation. The controlled wake has a narrow width and results in drag reduction. When the wind angle gradually increases, the stagnation point moves along the clockwise direction from point No. 1, and the mean pressure distribution becomes asymmetrical. When the wind angle increases from $15^{\circ}$ to $60^{\circ}$, the suction from the upper rotating cylinder is larger than that of the lower one. The mean negative pressure distribution of the lower rotating cylinder presents a small difference between $k = 2$ and $k = 0$, whereas the suction of the upper rotating cylinder at $k = 2$ has increased significantly. When $\theta = 90^{\circ}$, the difference of the peak negative pressures between two rotating cylinders has decreased, with a slightly lower negative pressure for the lower cylinder. As shown in Figure 15f–g, the stagnation point moves near the upper rotating cylinder, thus resulting in a reduction of the peak suction on the upper rotating cylinder. When $\theta = 180^{\circ}$, the stagnation zone is located in the region between the two rotating cylinders, and the mean pressure on the surface of the square cylinder presents a symmetrical distribution. The peak values of the negative pressures appear at points No. 9 and 25 in the flow separation zone.

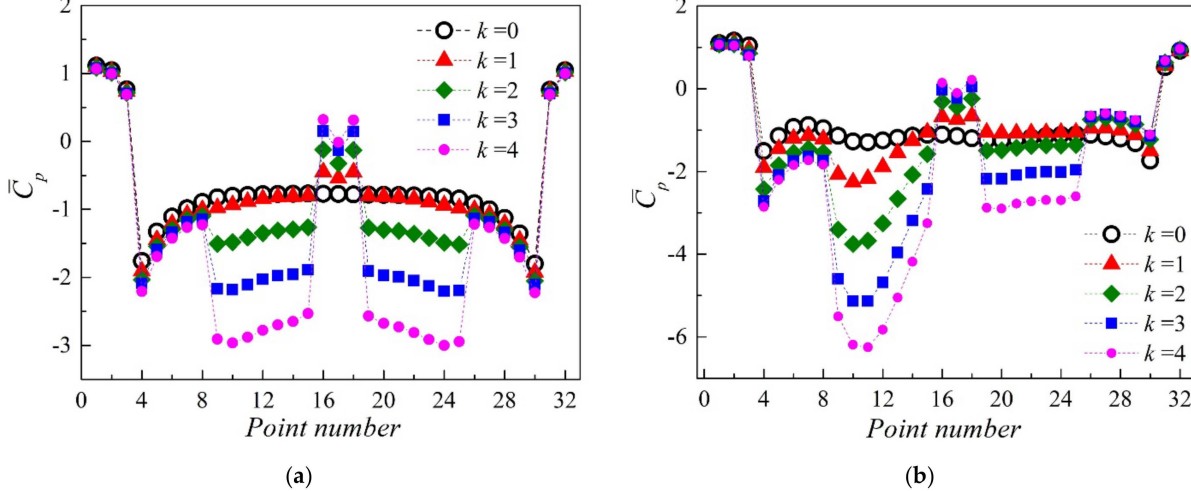

(a)

(b)

**Figure 15.** *Cont.*

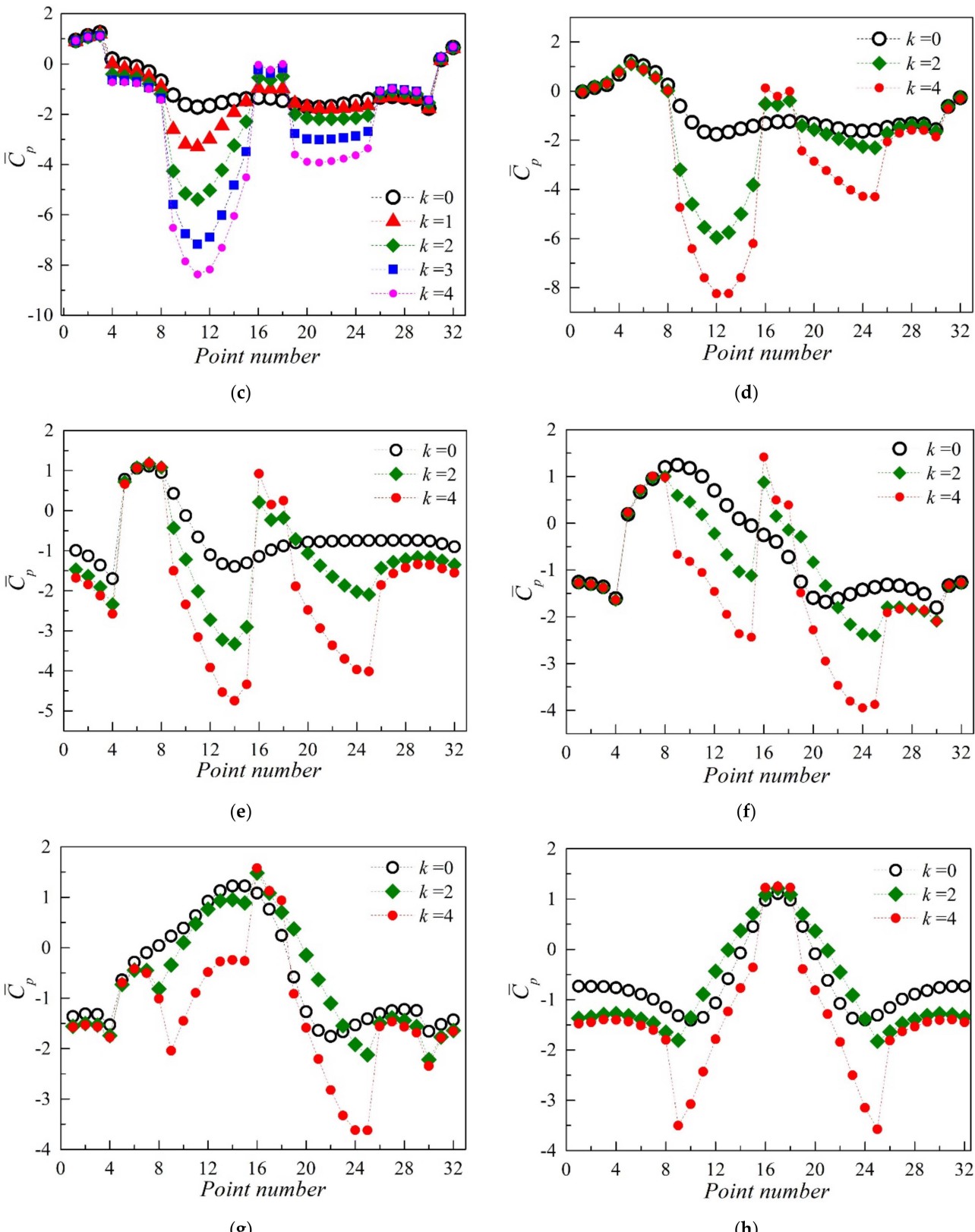

**Figure 15.** Mean pressure distribution varies with wind angles and velocity ratios (inward rotation), (**a**) $\theta = 0°$, (**b**) $\theta = 15°$, (**c**) $\theta = 30°$, (**d**) $\theta = 60°$, (**e**) $\theta = 90°$, (**f**) $\theta = 120°$, (**g**) $\theta = 150°$, (**h**) $\theta = 180°$.

Figure 16 shows the mean pressure distribution for different $k$ and $\theta$ values when the cylinder rotates outward. When $\theta = 0°$, the negative pressure of measuring points

No. 9–15 and No. 19–25 near the two small circular cylinders obviously increase. The mean pressure coefficient is asymmetrically distributed as the wind angle increases. The peak negative pressures of the lower rotating circular cylinder are higher than those of the upper circular cylinder when $\theta = 30° \sim 60°$. When $\theta = 90°$, the peak negative pressures of the two small rotation circular cylinders are similar; however, they are still asymmetrically distributed. When $\theta = 120° \sim 150°$, the stagnation point moves to the vicinity of the upper circular cylinder, thus resulting in a significant decrease in the negative pressure, whereas the negative pressure of the lower rotating circular cylinder increases significantly and reaches its maximum value at $\theta = 150°$. The stagnation point is located between the two rotating circular cylinders at $\theta = 180°$; the peak of the negative pressure appears in measuring points No. 10 and 24, and the mean pressure distribution on the surface of the square cylinder presents a symmetrical state, as well.

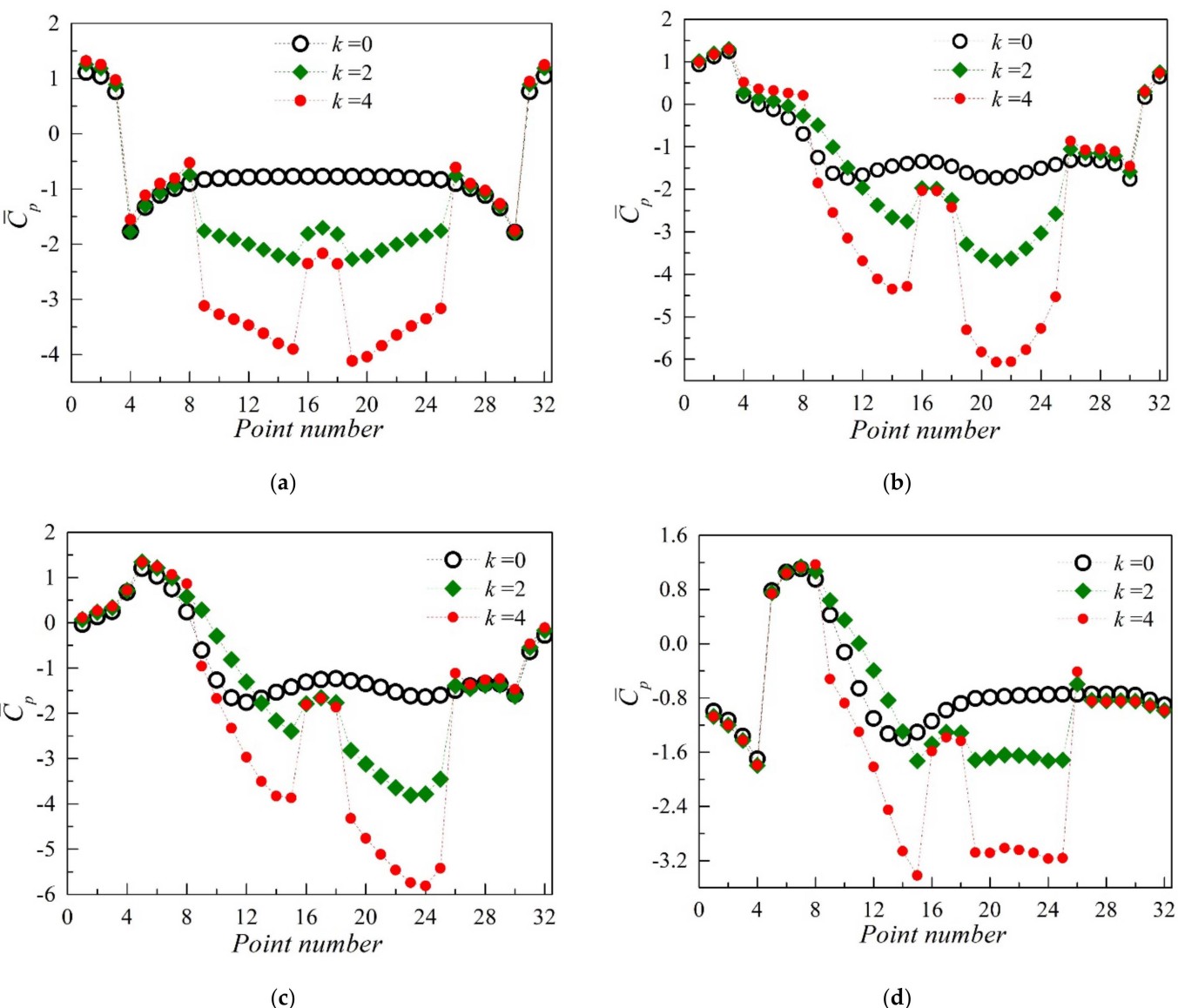

**Figure 16.** *Cont.*

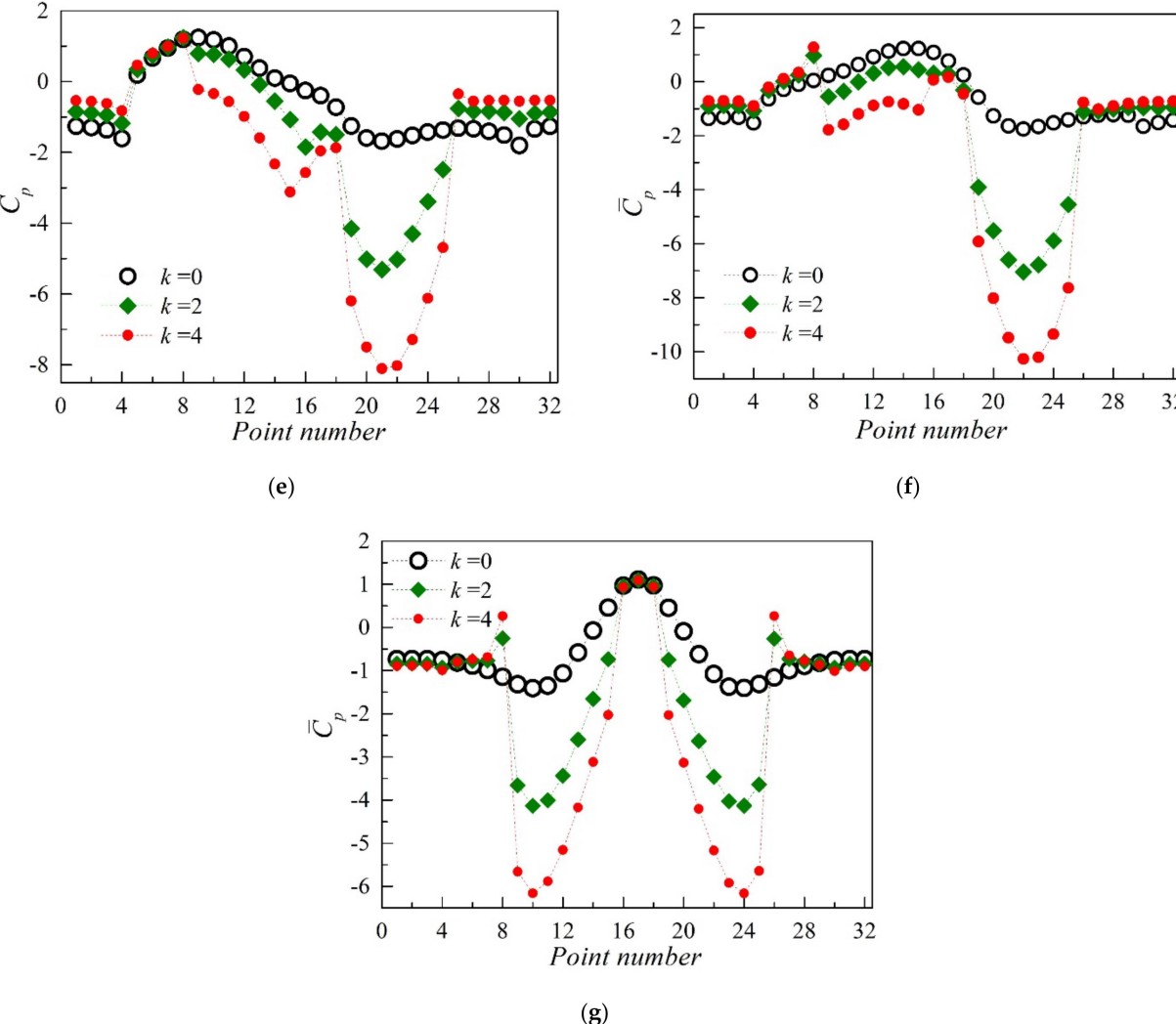

**Figure 16.** Mean pressure distribution varies with wind angles and velocity ratios (outward rotation), (**a**) $\theta = 0°$, (**b**) $\theta = 30°$, (**c**) $\theta = 60°$, (**d**) $\theta = 90°$, (**e**) $\theta = 120°$, (**f**) $\theta = 150°$, (**g**) $\theta = 180°$.

As shown in Figure 17a, the square cylinder position is symmetrical with respect to the incoming flow at $\theta = 0°$. However, the two small circular cylinders rotate with the same direction (co-rotation), resulting in an asymmetrical distribution of the mean pressure coefficient on the square cylinder surface and to a similar negative pressure on the measuring points near the small circular cylinders. The negative pressure around the upper rotating circular cylinder is significantly higher than that of the lower rotating circular cylinder at $\theta = 30° \sim 60°$. The upper circular cylinder is located at the flow separation zone at $\theta = 90°$, and the rotating direction is consistent with the flow acceleration direction, which can lead to a higher negative pressure on the upper circular cylinder. The analysis results in Figure 10 show that the wake vortex street of the square cylinder is eliminated in this particular wind angle. When $\theta = 120° \sim 150°$, the peak of the negative pressure on the surface of the upper cylinder gradually decreases, owing to the stagnation zone. While the lower circular cylinder is in the flow separation zone, the peak of the negative pressure significantly increases. When $\theta = 180°$, the rotating direction of the upper small circular cylinder is opposite to the flow direction of the separated shear layer, thus resulting in the decrease in the negative pressure on its surface. However, the rotating direction of the lower cylinder is consistent with the shear layer flow direction, which leads to an increment of negative pressure and to an asymmetrical distribution of the mean pressure.

Therefore, the fluctuation of the lift and drag coefficients increases when $k > 0$, as well, as shown in Figure 10.

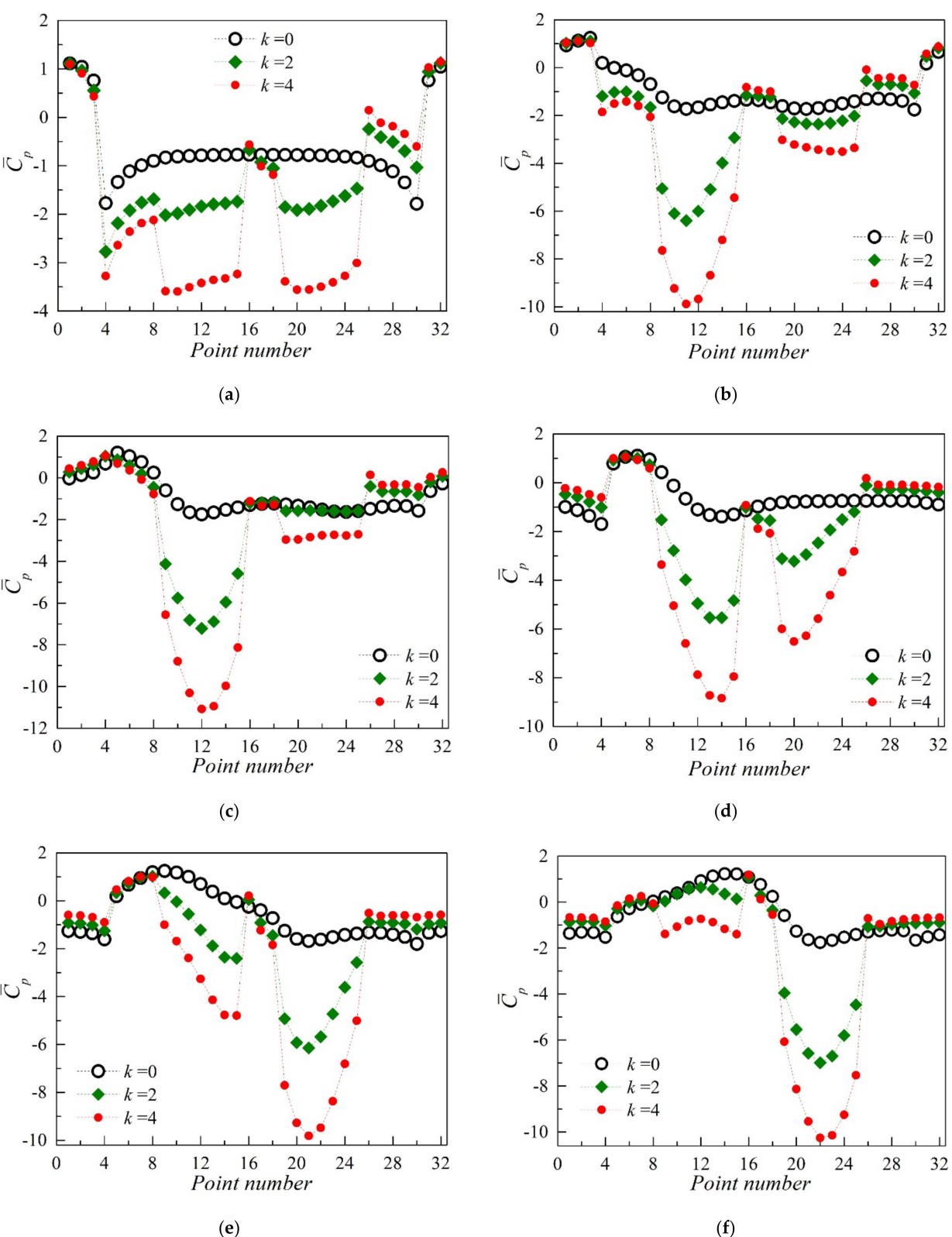

**Figure 17.** *Cont*.

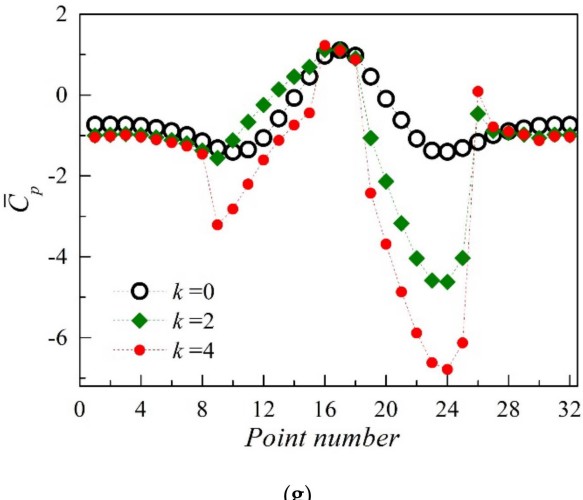

**(g)**

**Figure 17.** Mean pressure distribution varies with wind angles and velocity ratios (co-rotation), **(a)** $\theta = 0°$, **(b)** $\theta = 30°$, **(c)** $\theta = 60°$, **(d)** $\theta = 90°$, **(e)** $\theta = 120°$, **(f)** $\theta = 150°$, **(g)** $\theta = 180°$.

3.2.3. Aerodynamic Time History Analysis

This section presents the lift and drag coefficient evolutions of the square cylinder during the entire numerical simulation process, starting from the flow around the fixed square cylinder (simplified as 'FR') up to the MSBC stage. The time histories of the aerodynamic coefficients are compared before and after the rotation of the small circular cylinders. Figures 18 and 19 present the time histories of the lift and drag coefficients of the square cylinder for inward rotation and different $k$ values when the wind angle is $0°$ and $15°$, respectively. At these two wind angles, the MSBC plays an important role in suppressing the oscillating wake. In Figure 18a,b, each time history during the interval of $0 \le t < 100$ s indicates the lift and drag coefficients of the 'FR' stage. After the small circular cylinders have started to rotate at $t \ge 100$ s, the mean drag coefficient $\overline{C}_d$ decreases from 1.015 to 0.9, and the fluctuation of the lift coefficient decreases at $k = 1$, as well, as shown in Figure 18a; however, the vortex street in the wake of the square cylinder has not been completely eliminated in this case. With the increase in the rotating speed of the small circular cylinders, the input of momentum to the flow field gradually increases, as well. When $k = 2$, the fluctuation of the lift coefficient seems to have been eliminated, and $\overline{C}_d$ further decreases to 0.8. In Figure 18c,d, the MSBC was initialized from the fully developed flow at $t \ge 150$ s. When $k = 3$, the time history of the lift coefficient has the form of an almost straight line, without fluctuations, which indicates that the MSBC method completely eliminates the oscillating wake of the square cylinder. The $\overline{C}_d$ further decreases to 0.71, and the fluctuation of the drag coefficient decreases accordingly. When $k = 4$, $\overline{C}_d$ continues to decrease to 0.65, which indicates that the drag coefficient decreases as $k$ increases; however, the fluctuation of the lift and drag coefficient is larger than that for $k = 3$. This suggests that the case of $k = 2$ can be regarded as the optimal choice of the MSBC method, in terms of the control effect and the external energy consumed.

When $t \ge 200$ s, the square cylinder activates the MSBC from the fully developed flow when the wind angle is $15°$, as shown in Figure 19. With the increase in $k$ from 1 to 4, the fluctuation value $C_l\prime$ of the lift coefficient decreases significantly, whereas $\overline{C}_l$ increases from 0.29 to 1.54. The drag coefficient $\overline{C}_d$ decreases to 0.87 at $k = 4$, which corresponds to a 54.2% decrease compared to of its value when $k = 0$. It may be observed that the fluctuation of the lift and drag coefficients decreases with the increase in the input of momentum, whereas the mean of the lift coefficient increases, and the mean of the drag coefficient decreases. The MSBC plays an important role in the decrease in the mean of the drag force and in the increment of the mean lift force.

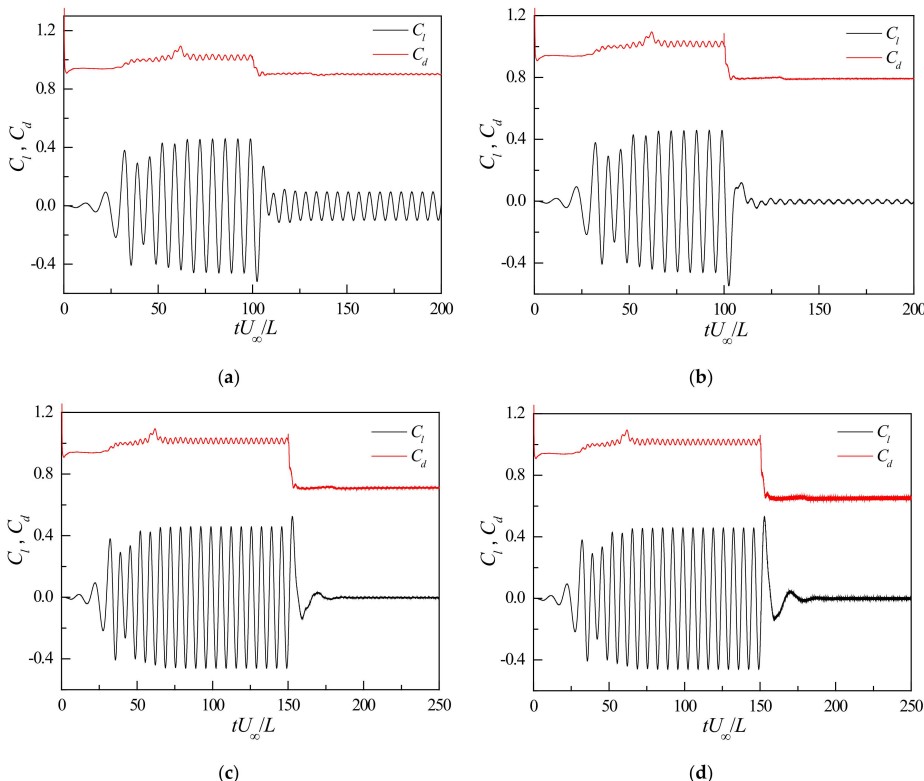

**Figure 18.** Lift and drag coefficient time histories of the square cylinder with different velocity ratios at $\theta = 0^\circ$ (inward rotation), (**a**) $k = 1$, (**b**) $k = 2$, (**c**) $k = 3$, (**d**) $k = 4$.

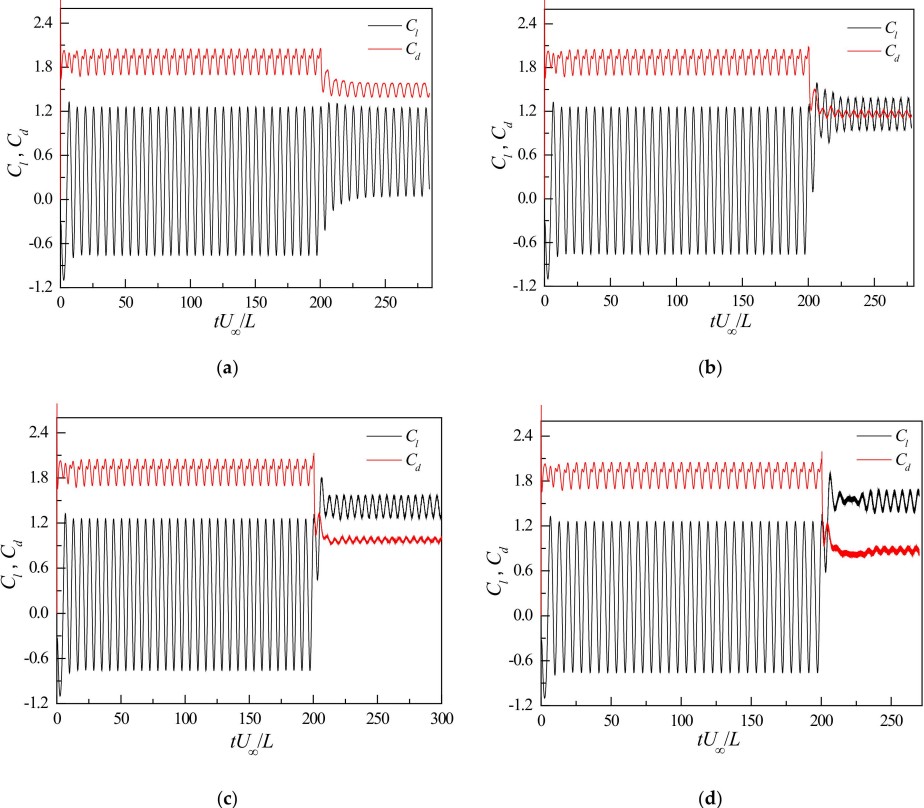

**Figure 19.** Lift and drag coefficient time histories of the square cylinder with different velocity ratios at $\theta = 15^\circ$ (inward rotation), (**a**) $k = 1$, (**b**) $k = 2$, (**c**) $k = 3$, (**d**) $k = 4$.

Figure 20 shows the comparative analysis of time histories of the lift and drag coefficients for inward rotation and $\theta = 0°$ when MSBC becomes activated at different times. The simulation is divided into two cases. In the first case, the small circular cylinders are not rotating, and the square cylinder is at the 'FR' stage at $0 \leq t < 100$ s; when the alternating vortex shedding was steady; the MSBC became activated at $t = 100$ s with $k = 1.5$. In the second case, the MSBC becomes activated for $k = 1.5$ to inject the momentum into the flow field from $t = 0$ s. Although the activation time of the MSBC is different, the two cases have the same control effect on the lift and drag forces of the square cylinder. Only a small phase difference can be found in the comparison of the lift coefficient.

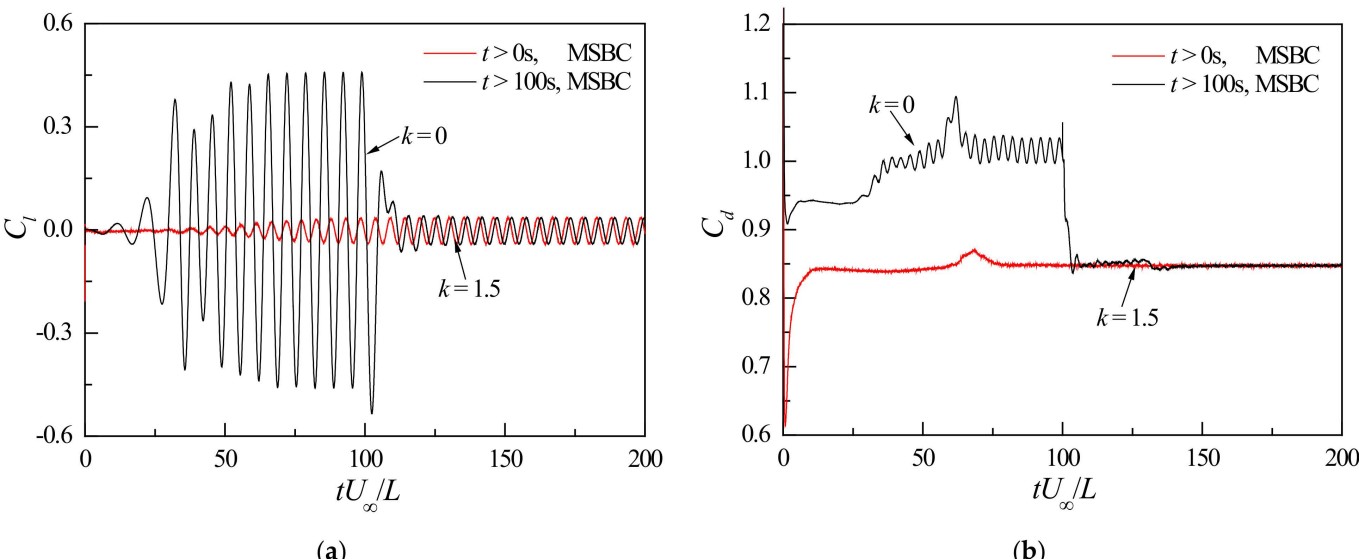

**Figure 20.** Comparisons of lift and drag coefficient time histories of the square cylinder while starting MSBC at different time under $\theta = 0°$ and inward rotation, (**a**) $C_l$, (**b**) $C_d$.

Figures 21 and 22 show the lift and drag coefficient evolutions of the square cylinder for outward rotation and different $k$ values when the wind angle is $30°$ and $120°$, respectively. In Figure 21, it may be observed that the fluctuation of the lift coefficient increases with the increase in $k$ when $\theta = 30°$ and $t \geq 200$ s; the mean and the fluctuation of the drag coefficient increases with the increase in $k$, as well. This indicates that the input of momentum that has been generated by the MSBC method enhances the vortex shedding energy in the wake of the square cylinder. Figure 22 shows that the MSBC has an effect of 'drag reduction and lift increment' at $\theta = 120°$. The mean and the fluctuation of the drag coefficient decrease with the increase in the value of $k$ after the small circular cylinders start to rotate. Simultaneously, the mean of the lift coefficient increases with the increase in $k$, and the fluctuation of the lift coefficient decreases with the increase in $k$, although it still exists when $k = 4$. This shows that the wake vortex shedding of the square cylinder has not been completely eliminated; however, the eddy energy has decreased significantly. Similar conclusions can be drawn from the peak value of the amplitude spectra in Figure 12c.

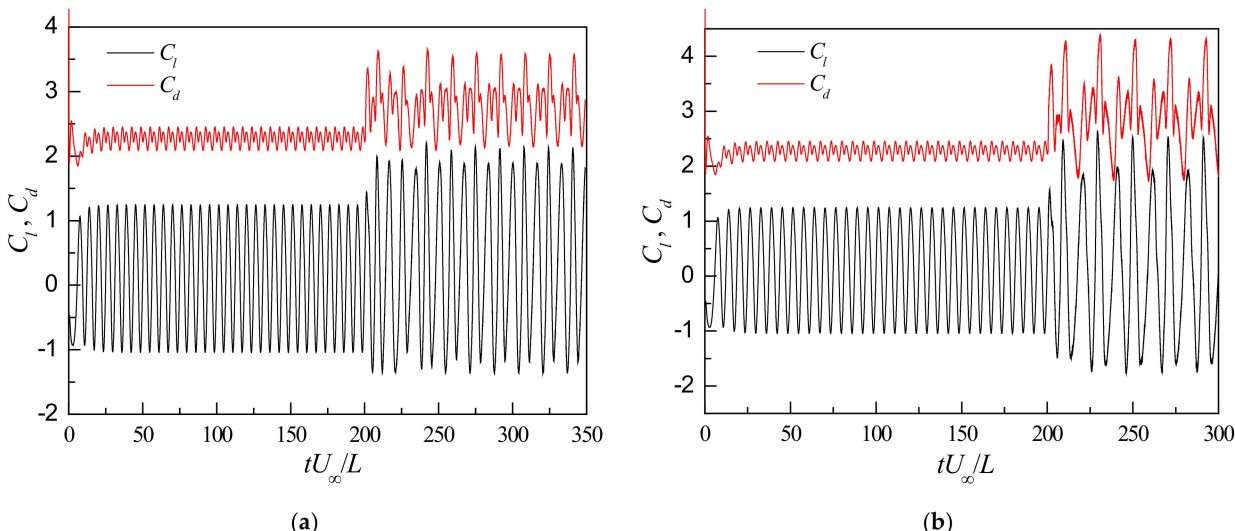

**Figure 21.** Lift and drag coefficient time histories of the square cylinder with different velocity ratios at $\theta = 30°$ (outward rotation), (**a**) $k = 2$, (**b**) $k = 4$.

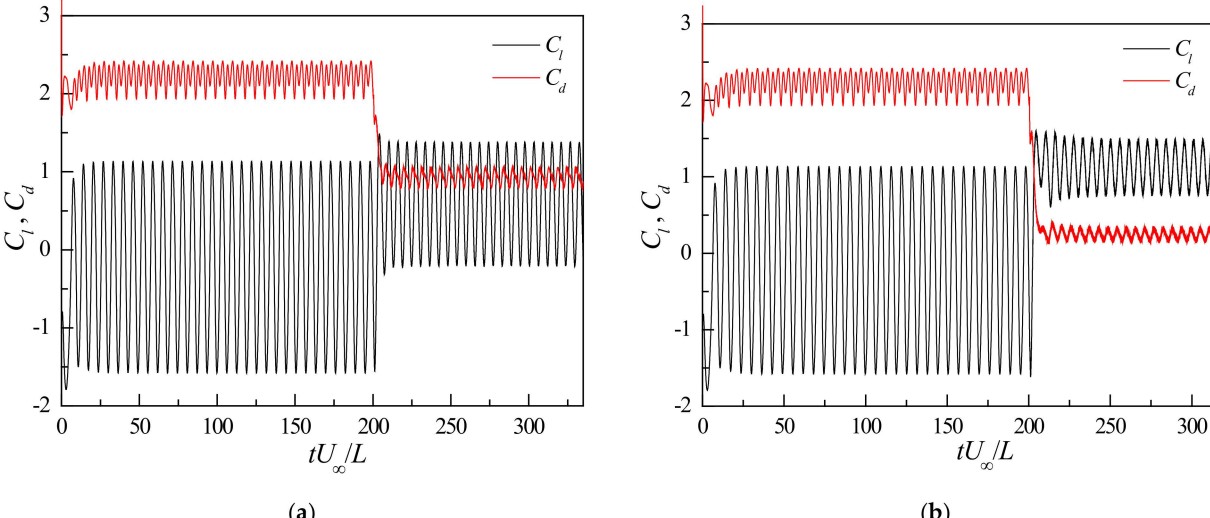

**Figure 22.** Lift and drag coefficient time histories of the square cylinder with different velocity ratios at $\theta = 120°$ (outward rotation), (**a**) $k = 2$, (**b**) $k = 4$.

Figures 23 and 24 show the time histories of the lift and drag coefficients of the square cylinder with clockwise co-rotation and different $k$ values when the wind angle is $60°$ and $90°$, respectively. As shown in Figure 23, the mean and the fluctuation of the drag coefficient significantly decrease; the mean of the lift coefficient increases, and the fluctuation level is still high when $\theta = 60°$ and $k = 2$. This indicates that the wake vortex shedding of the square cylinder is still in existence. With the increase in the momentum input, the mean of the drag coefficient continues to decrease, and the mean of the lift coefficient increases when $k = 4$; however, the fluctuation has been significantly reduced. This suggests that the wake vortex street of the square cylinder was suppressed; however, it was not entirely eliminated. As shown in Figure 24, as $k$ further increases, the mean and the fluctuation of the drag coefficient of the square cylinder present a small change; however, the mean of the lift coefficient increases gradually, and the fluctuation of the lift coefficient significantly decreases when $\theta = 90°$. This suggests that the oscillating wake of the square cylinder is effectively suppressed. High-frequency parts of the lift and drag coefficients are

irrelevant to the vortex frequency, which corresponds to the rotation frequency of the small circular cylinders.

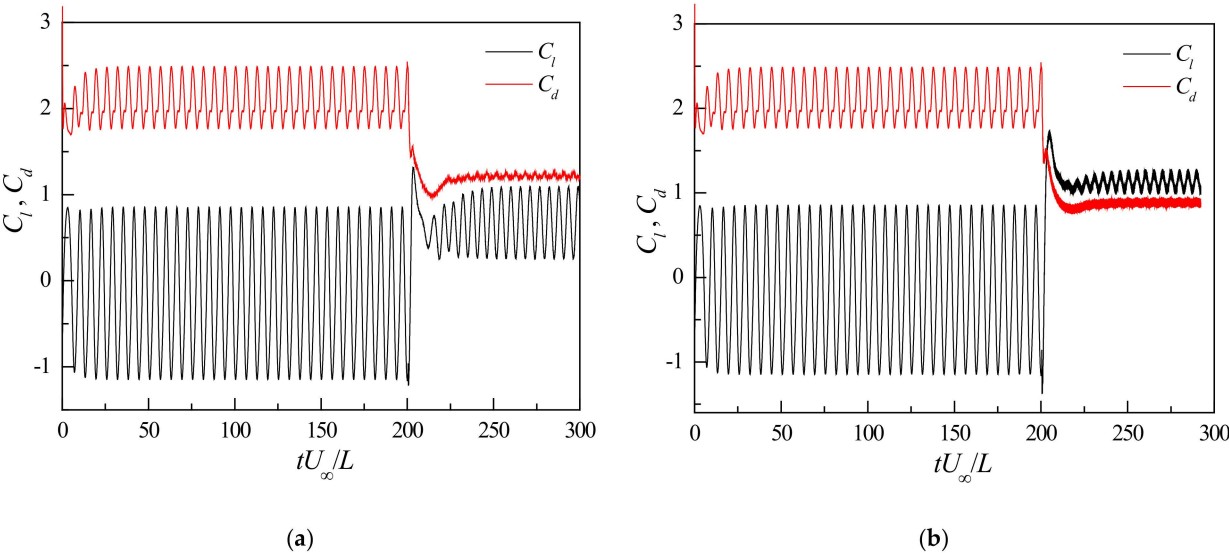

**Figure 23.** Lift and drag coefficient time histories of the square cylinder with different velocity ratios at $\theta = 60^{\circ}$ (co-rotation), (**a**) $k = 2$, (**b**) $k = 4$.

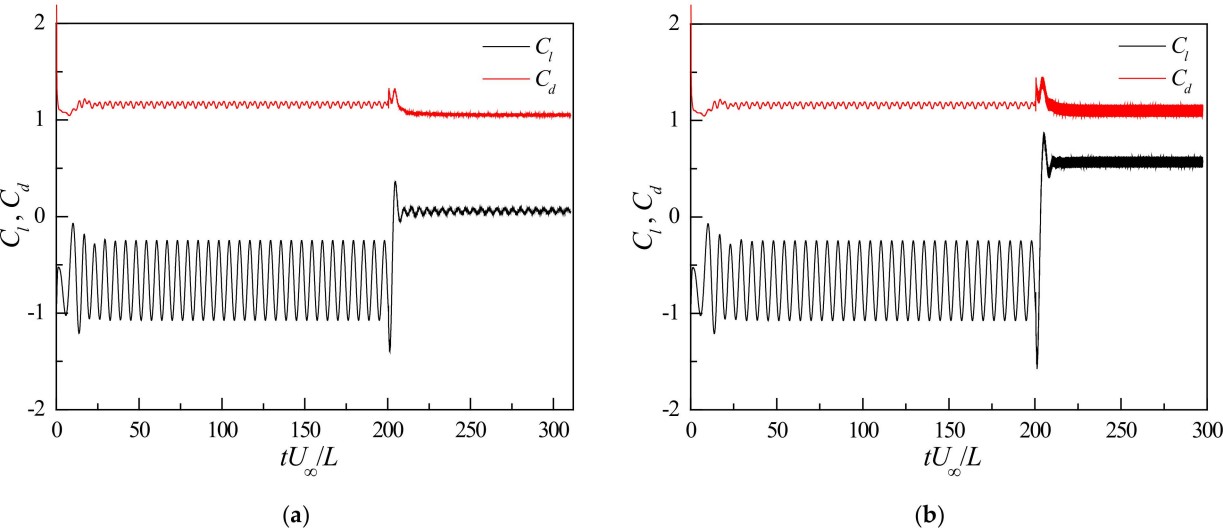

**Figure 24.** Lift and drag coefficient time histories of the square cylinder with different velocity ratios at $\theta = 90^{\circ}$ (co-rotation), (**a**) $k = 2$, (**b**) $k = 4$.

### 3.2.4. Wake Vortex Structure

The momentum injection of the MSBC method affects the vortex characteristics of the flow field, and the different vortex structures and shedding modes are directly related to the aerodynamic forces of the square cylinder. In order to demonstrate the control effect of the MSBC method on the flow field around the square cylinder, in this study, we present the vortex wake of the square cylinder with representative wind angles for different rotation modes. Figure 25 shows the vorticity contours of the flow field around the square cylinder at different $k$ values for inward rotation and wind angles of $0^{\circ}$ and $15^{\circ}$. In Figure 25a, when $k = 0$, the shear layers separate from the leading edge corner of the square cylinder and enter into the wake region; then, they immediately curl and develop into an alternating shedding vortex street, which causes the fluctuation of the lift and drag coefficients. When $k = 1$, the rotating cylinder begins to transfer momentum into the flow field; however, the

wake vortex street still exists. Two quasi-stable small vortex structures are formed near the two small circular cylinders at the leeward side of the square cylinder, and they lead to the shear layer separating from the leading edge corner, which rolls up to a slightly farther downstream location. This reduces the lift coefficient by 80%. As the $k$ value increases, the separated shear layer rolls up farther downstream, and the vorticity intensity and scale slightly decrease. Two small symmetric stable vortices on the leeward side of the square cylinder have gradually stretched, thus preventing the interaction between the upper and lower shear layers. When $k = 4$, the large-scale alternate shedding vortex street in the square cylinder wake has been completely eliminated by the small-scale vortices induced by the two rotating cylinders, and the shear layer has not rolled up in the computational domain again. Obviously, for this rotation direction and wind angle, the momentum injected into the flow field by the small cylinders delays the separation of the shear layer and narrows the wake region. It finally eliminates the wake vortex street and forms a symmetrical stable vortex structure in the wake of a steady flow. Figure 25b shows that the vortex near the square cylinder wake is stable and develops into the alternate shedding vortex slightly farther downstream with the increase in $k$ when $\theta = 15^\circ$. The fluctuation of the lift coefficient is significantly reduced, as well; however, the vortex street has not been completely eliminated at $k = 4$. The vortex street presents an inclined downward direction, which is the main cause of the increase in the mean lift coefficient. This is consistent with the results shown in Figure 8a.

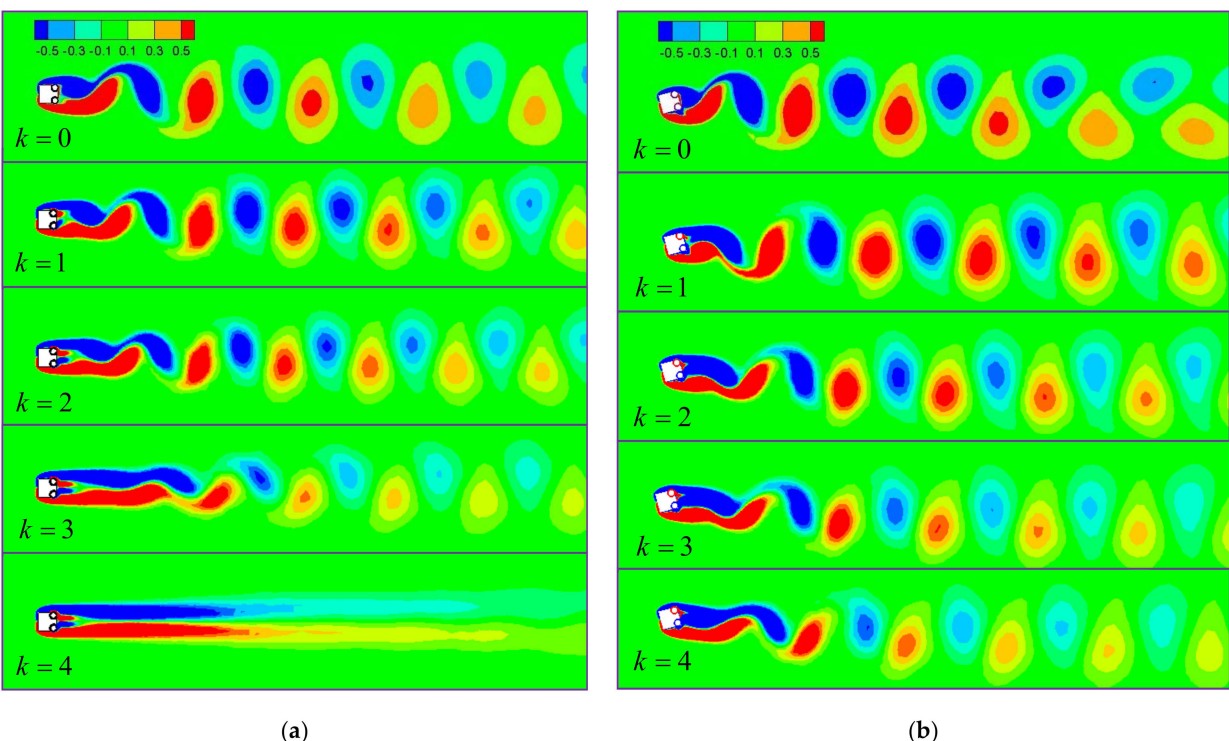

(**a**)                                                     (**b**)

**Figure 25.** Vorticity contours in the wake of the square cylinder under inward rotation mode, (**a**) $\theta = 0^\circ$, (**b**) $\theta = 15^\circ$.

For the case of inward rotation mode and $\theta = 0^\circ$, the velocity contours and streamlines in the near wake of the square cylinder with different velocity ratios are shown in Figure 26, which can clearly show the close-up view around the region of rotating cylinders. In order to compare the results of different cases more clearly, the upper limit value of the velocity in the legend of each subgraph is consistent, which is the maximum speed in the flow field when $k = 0$.

In Figure 26a, a large-scale vortex is generated in the upper region of the near wake immediately adjacent to the leeward side of the square cylinder, and the lift coefficient of

the square cylinder reaches a positive maximum value at this time. The streamline at the downstream of the main vortex is obviously curved, indicating that the Karman vortex street with alternating shedding exists in the wake of the square cylinder. When $k = 1$, it can be seen from Figure 26b that the flow velocity around the small cylinders is increased, and the rotating cylinders started to inject momentum into the wake. At this time, there are still shedding main vortices and obviously curved streamlines in the square cylinder wake. Compared with Figure 26a, the low-velocity region in the wake begins to narrow, the scale of the main vortex is decreased, and the distance between the vortex center and the square cylinder is increased. The influence of the main vortex on the square cylinder begins to weaken, and the fluctuation of the lift coefficient is reduced to a certain extent, which is consistent with the result in Figure 18a. When $k = 2$, $3$, as the velocity ratio increases gradually, the velocity around the small cylinders is increased gradually, and the momentum injected into the wake field is also increased gradually, as shown in Figure 26c,d. Meanwhile, two symmetrical small vortices appear in the near wake region of the square cylinder, and the alternating shedding main vortices are pushed farther downstream. The scale and energy of the main vortices are gradually decreased, and the fluctuation of the corresponding lift coefficient is also reduced significantly, as shown in Figure 18b,c.

As shown in Figure 26e, the small circle cylinders rotating with high velocity ratio of $k = 4$ inject more momentum to the wake. The high momentum is mainly concentrated in the near wake range of one time of the side length of the square cylinder and symmetrically distributed on both sides of the centerline of the square cylinder wake. There is a pair of stable and symmetrical small vortices between the high-momentum region and the upper and lower shear layers separated from the corners of the leading edge of the square cylinder. The low-velocity region of the square cylinder wake becomes narrower, the separated shear layers are no longer curled, and the main vortex have entirely disappeared. The streamlines are symmetrically distributed on both sides of the wake centerline and became more and more straight as they move downstream. At this time, the fluctuation of the lift coefficient has disappeared, and the mean value of the drag coefficient is also decreased to its minimum. It can be seen that MSBC achieves the purpose of completely eliminating the alternating shedding vortices by inputting high momentum into the wake of the square cylinder.

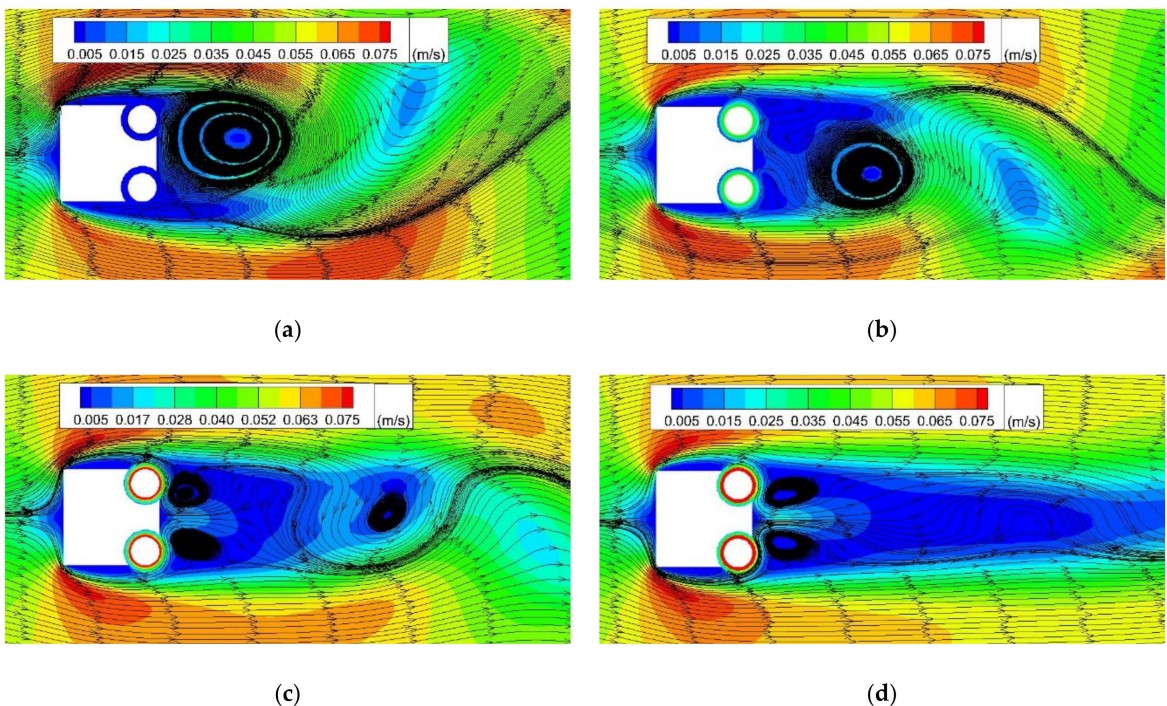

(a)

(b)

(c)

(d)

**Figure 26.** *Cont.*

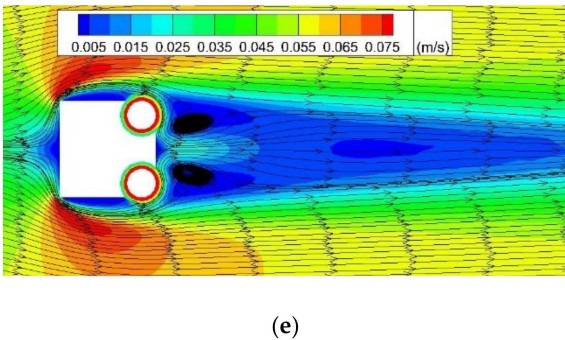

(**e**)

**Figure 26.** Velocity contours and streamlines in the near wake of the square cylinder under inward rotation mode and $\theta = 0^\circ$, (**a**) $k = 0$, (**b**) $k = 1$, (**c**) $k = 2$, (**d**) $k = 3$, (**e**) $k = 4$.

Figure 27 shows the vorticity contours of the flow field around the square cylinder for different $k$ values, with an outward rotation, and for wind angles of $30^\circ$ and $120^\circ$. The momentum injected by the rotating cylinders significantly influences the wake field at $\theta = 30^\circ$. When $k = 2$ and $k = 4$, vortices of different scales appear in the flow field, and the vortex shedding patterns are very complex. As shown in Figure 27a, the vortex shedding pattern is obviously different from the regular vortex shedding model when $k = 0$; the drastic change in the shedding mode and the vortex scale cause a significant increase in the lift and drag coefficients. As shown in Figure 27b, the two small rotation cylinders on the upper side of the square cylinder cannot control the shear layer that has separated from lower side. When $k = 4$, the vortex street still exists, and the entire vortex street presents an inclined downward direction. This leads to a positive mean value of the lift coefficient, which increases with the increase in $k$.

Figure 28 shows the vorticity contours of the flow field around the square cylinder for different $k$ values, with clockwise co-rotation, and for wind angles of $60^\circ$ and $90^\circ$. When $\theta = 60^\circ$, the evolution process of the flow field around the square cylinder for various $k$ values is similar to that of Figure 27b, and the vortex shedding mode presents an effect of 'drag reduction and lift increment' at this moment. The vortex near the square cylinder wake is stable, and the vortex street is formed farther downstream, when $k = 4$. Therefore, the fluctuation of the lift coefficient is significantly reduced, as well. Figure 28b shows that when the small circular cylinders do not rotate, the stable '2S' mode is formed in the square cylinder wake. The shear layer near square cylinder wake is stable and rolls up to a slightly farther downstream location when $k = 2$. At this time, the vortex has not been eliminated; however, the fluctuation value of the lift coefficient has decreased by 90% compared with that when $k = 0$. When $k = 4$, the mean of the lift coefficient is positive, and it increases with the increase in $k$ value because of the overall downward deflection of the steady wake. At this point, the alternating shedding vortex in the wake of the square cylinder has been completely eliminated because the two small circular cylinders are both on the upper side of the square cylinder, and they rotate downstream together; this has a strong suction effect on the upper shear layer. Therefore, the shear layer that separated from lower side of the square cylinder is unable to curl into a vortex, and the wake presents an overall downward deflection, which leads to the increase in the mean of the lift coefficient. This is a newly discovered control mode, that is, when rotating cylinders are arranged on one side of the structure, the control effectiveness can also be achieved. For example, only one side of a long-span bridge deck is used for vehicle traffic, and the other side can be equipped with multiple rotating devices, such as vertical axis wind turbines placed horizontally, which can control the wake of the bridge deck while collecting wind energy.

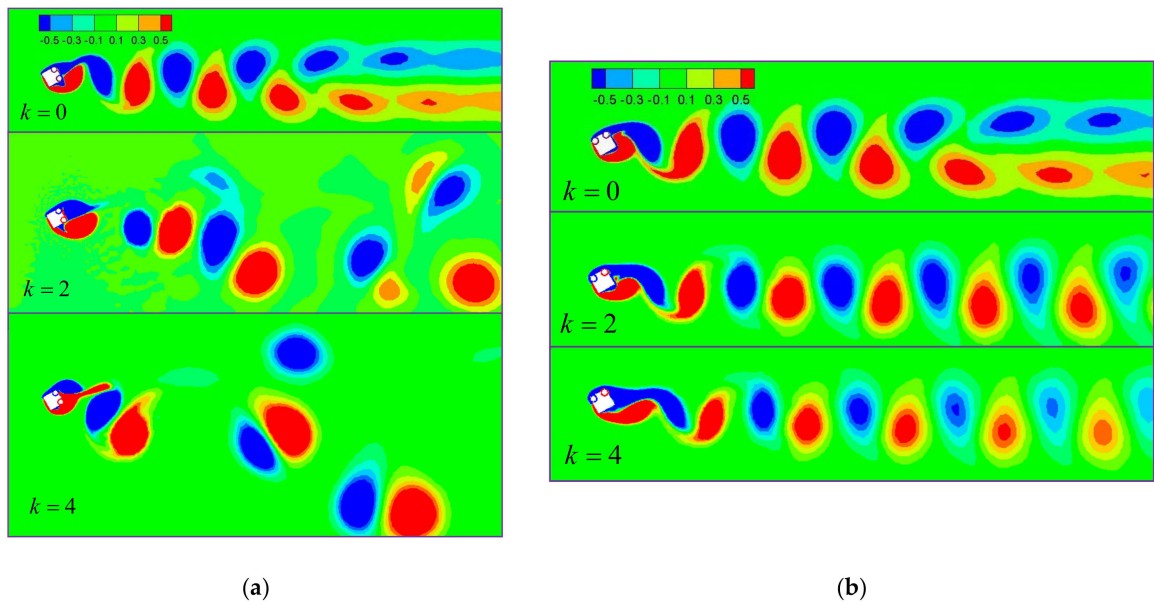

**Figure 27.** Vorticity contours in the wake of the square cylinder under outward rotation mode, (**a**) $\theta = 30^{\circ}$, (**b**) $\theta = 120^{\circ}$.

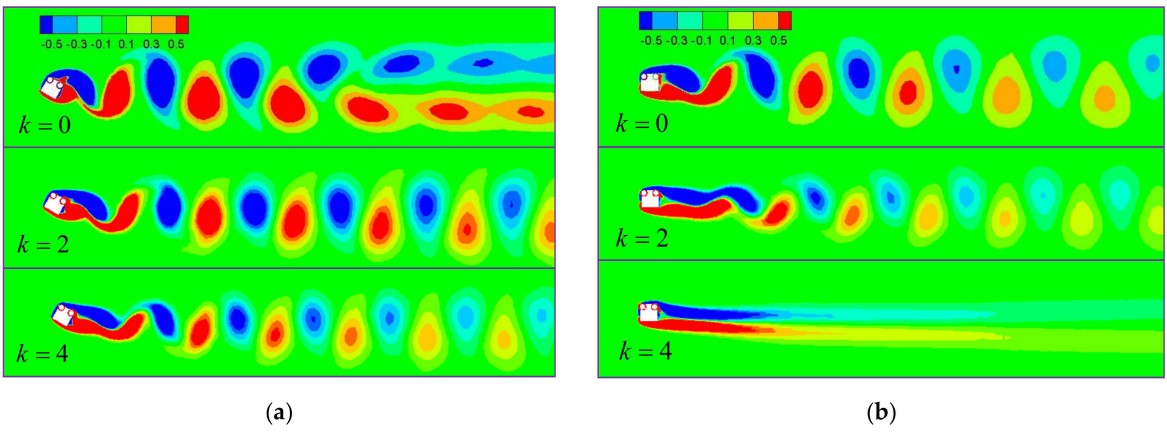

**Figure 28.** Vorticity contours in the wake of the square cylinder under co-rotation mode, (**a**) $\theta = 60^{\circ}$, (**b**) $\theta = 90^{\circ}$.

## 4. Conclusions

In this study, the computational fluid dynamics (CFD) numerical simulation was employed for the study of the MSBC effect on the oscillating wake of a square cylinder. We conducted a numerical simulation of the entire process, from the flow around a fixed square cylinder to the oscillating wake controlled via the MSBC, and the conclusions obtained from the study are as follows:

For the inward rotation mode and wind angles of $\theta < 60^{\circ}$, the MSBC method plays an important role in suppressing the oscillating wake of the square cylinder; the effectiveness of the control in suppressing the wake was more obvious when the wind angle was smaller. In the other wind direction, the MSBC method was detrimental to the wake stability of the square cylinder. The fluctuation value of the lift coefficient decreased by 98%, and the mean value of the drag coefficient decreased by 36% when $\theta = 0^{\circ}$ and $k = 4$. In this case, the elimination of the oscillation wake of the square cylinder produced the best effect. In terms of control effectiveness and the external energy consumed, the velocity ratio $k = 2$ can be regarded as the optimal choice of the MSBC control method at Re = 200.

For the outward rotation mode and when $\theta = 120^{\circ}$ and $150^{\circ}$, the effect of suppressing oscillating wake of the square cylinder continuously improved with the increase in $k$. In

other wind directions, the momentum input that was generated by the rotating cylinders plays an important role in enhancing the wake of the square cylinder.

For the clockwise co-rotation and when $\theta = 0^{\circ}$ and $180^{\circ}$, the momentum input of the rotating circular cylinders was unfavorable to the wake stability of the square cylinder. In other wind directions, the oscillating wake of the square cylinder was suppressed to different levels. Among them, the effect of eliminating the oscillation wake of the square cylinder was most obvious for wind angles of $60^{\circ}$ and $90^{\circ}$.

The high suction region on the surface of the rotating circular cylinders can prevent streamline bending, delay the separation of the shear layer, and form a narrow wake region, which transforms the unsteady flow in the wake of the square cylinder into an approximately steady flow. This leads to the decrease in the mean of the drag coefficient and the fluctuation of the lift coefficient; thus, suppressing the flow-induced vibration of the square cylinder can be achieved.

**Author Contributions:** Conceptualization, T.S. and F.X.; methodology, T.S. and F.X.; software, X.L.; validation, T.S. and F.X.; formal analysis, F.X.; investigation, T.S. and X.L.; resources, F.X.; data curation, X.L.; writing—original draft preparation, T.S.; writing—review and editing, F.X.; visualization, X.L.; supervision, F.X.; project administration, F.X.; funding acquisition, F.X. All authors have read and agreed to the published version of the manuscript.

**Funding:** This research was funded by the National Natural Sciences Foundation of China (NSFC) (52078175, 51778199 and U1709207), the Natural Science Foundation of Guangdong Province (2019A15 15012205), the fundamental research funds of Shenzhen Science and Technology plan (JCYJ2019080614 4009332, JCYJ201803306172123896), and the Stability Support Program for colleges and universities in Shenzhen (GXWD20201230155427003-20200823134428001).

**Data Availability Statement:** Data are contained within this article.

**Conflicts of Interest:** The authors declare no conflict of interest.

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
