# Peer review of "Moving Surface Boundary-Layer Control on the Wake of Flow around a Square Cylinder"

_applsci, doi:10.3390/app12031632_

Round 1

Reviewer 1 Report

The manuscript presented by the authors illustrates the process of the flow around a fixed square cylinder and the moving surface boundary-layer control at a low Reynolds number by performing numerical simulations. As a reviewer, I believe that the work is interesting.  Therefore, I think it should eventually be published. However, while reading the manuscript this reviewer found a few elements that would benefit of clarification, further justification, or just change. These few elements are pointed next through a short detailed list. 

The Introduction is comprehensive and balanced. 

1- Readers would benefit from more information on how the equations that govern the flow are solved. In this aspect, how is pressure solved? What is the typical computational time for each round? 

2- Could be useful for the readers that the authors provide more details about the mesh-grid used and the number of mesh-grid points.

3 - How long is each simulation run? Is there any steady-state for calculating statistics? If there is how much is it reached? In this aspect, please provide more details on how the statistics were calculated.

4- Line 208 - U_inf / L:  please, provide the typical values for U_inf and L. 

5- What represent the different color in Fig. 13?

6- What is "?" in the caption of Fig. 14. (a) ?

7- Fig. 25-27: Please add what represents the colors in the plot or provide more information on the caption. 

Author Response

Response to Reviewer 1 Comments

 applsci-1472032, Applied Sciences

Title: Moving surface boundary-layer control on the wake of flow around a square cylinder

Authors: Te Song, Xin Liu, Feng Xu

The insightful comments and helpful suggestions given by the reviewers are greatly appreciated. In response to the reviewers’ comments, this document is prepared and submitted together with the revised manuscript.

In the context below, the words in black are taken from the reviewer’s comments. Our responses are given in the sections that follow in red color, and the modifications in the revised manuscript according to the reviewer’s comments are marked in red color.

~~~~~~~~~~~~~~~~~~~~~~~~~~~~~~~~~~~~~~~~~~~~~~~~~~~

Reviewers' comments and Responses:

Reviewer #1:

The manuscript presented by the authors illustrates the process of the flow around a fixed square cylinder and the moving surface boundary-layer control at a low Reynolds number by performing numerical simulations. As a reviewer, I believe that the work is interesting.  Therefore, I think it should eventually be published. However, while reading the manuscript this reviewer found a few elements that would benefit of clarification, further justification, or just change. These few elements are pointed next through a short detailed list. 

The Introduction is comprehensive and balanced. 

Point 1: Readers would benefit from more information on how the equations that govern the flow are solved. In this aspect, how is pressure solved? What is the typical computational time for each round? 

Response 1: The governing equations of fluid flow include continuity equation and momentum (N-S) equation. The solution of the governing equations involves the problems of pressure-velocity and coupling. This is because each velocity component appears in both the momentum equation and the continuity equation, causing the equations to couple together. At the same time, the pressure term also appears in the momentum equation, but there is no equation that can be directly solved for the pressure. The Fluent software includes three pressure-velocity coupling solving algorithms, SIMPLE, SIMPLEC and PISO, of which the SIMPLE (Semi-Implicit Method for Pressure-Linked Equations) algorithm is the default and robust scheme. The basic idea of the SIMPLE algorithm is to give the pressure field (the assumed value, or the result of the last iteration), and obtain the velocity field by solving the discrete momentum equation. Because the pressure field is assumed or imprecise, the obtained velocity field generally does not satisfy the continuity equation. Therefore, the given pressure field must be corrected. The principle of correction is that the velocity field corresponding to the corrected pressure field can satisfy the continuity equation. According to this principle, the relationship between pressure and velocity specified by the momentum equation is brought into the continuity equation to obtain the pressure correction equation, and the pressure correction value is calculated from this equation. Next, based on the corrected pressure field, a new velocity field is obtained, and then it is checked whether the velocity field converges. The SIMPLE algorithm used in this study has been described in the original manuscript as detailed in the red text on page 6, lines 199-201.

The present numerical study is carried out at Re=200, the fluid medium is air, the side length L of the square column is 0.054 m, and the incoming velocity ­U is 0.054 m/s. The abscissa of the aerodynamic coefficient time-history curves given in the manuscript is the dimensionless time t*=tU/L. The actual computation time t=t*L/U= t*. For the uncontrolled flow around the square cylinder, as shown in Fig. 4, the calculation time lasts until the alternating shedding vortices appear in the wake (150 s), the amplitude of the corresponding lift and drag coefficient is stable, and it can be ensured that there is a sufficient length of stable aerodynamic time history results for statistical analysis (80~150 s). For the controlled case of flow around a square cylinder, the total calculation time includes two parts, including the calculation time of the uncontrolled flow around square cylinder stage and the calculation time of the controlled stage. The calculation starts at t=0. When the wake vortex shedding is stable, the small cylinder is started to perform the flow control calculation, and the calculation ends when the alternating shedding vortex of the wake disappears or the aerodynamic amplitude of the controlled cylinder is stable. As shown in Fig. 21, the calculation time of the uncontrolled flow around square cylinder stage is 200 s, and the calculation time of the controlled stage is 150 s and 100 s, respectively. It can be seen that the amplitudes of the aerodynamic coefficients at each stage have entered a stable stage. Since the parameters such as wind direction angle and rotational speed of small cylinder are different in different cases, the calculation time of the controlled stage is not necessarily the same. The calculation is stopped and the total calculation time is determined according to the actual control effect obtained.

Point 2: Could be useful for the readers that the authors provide more details about the mesh-grid used and the number of mesh-grid points.

Response 2: For the various meshing schemes of the single square cylinder shown in Table 1, the minimum grid size of each edge of the square cylinder is equal and both are lmin=L/Nc. The grid grows into the computational domain with each edge of the square cylinder as the source and a growth ratio of 1.04, and the maximum grid size lmax=0.4L. The total number of grid nodes in the computational domain has been supplemented in Table 1. Corresponding text descriptions have been added to the revised manuscript, as detailed in red text on page 6, lines 213-216.

For the case of controlled square cylinder, the minimum grid size of the square cylinder surface (four straight edges and two arcs) and the two rotating small cylinders surfaces (shown by the red lines in Fig. 2(b)) are the same, and are equal to the minimum grid size lmin=L/70 for the single square cylinder 'Scheme 3' grid scheme. The grid grows into the computational domain with the square cylinder surface and the two rotating small cylinders surfaces as the source and the growth ratio of 1.04, and the maximum grid size is also lmax=0.4L.

The corresponding note has been added to the red text in lines 227-229 on page 6 of the revised manuscript.

Point 3: How long is each simulation run? Is there any steady-state for calculating statistics? If there is how much is it reached? In this aspect, please provide more details on how the statistics were calculated.

Response 3: The serial or one process of the Intel i5 CPU was used for the calculation in this study, and it took about 4~5 hours to complete the calculation of the whole process from flow around square cylinder to flow control. Due to the continuous improvement of computer hardware technology, if a higher performance CPU or parallel computing is used, the time required to complete a simulation will be significantly reduced.

The amplitudes of the aerodynamic force and pressure time-history curves calculated under all cases in this study reached a stable state, and the aerodynamic force and pressure results of the time period with stable amplitude were used for statistical analysis. Usually, the time period with stable amplitude reached at least 5 characteristic period of vortex shedding.

The corresponding text descriptions have been added to the revised manuscript as detailed in the red text on page 7, lines 238-242.

Point 4: Line 208 - U_inf / L:  please, provide the typical values for U_inf and L.

Response 4: The typical values of U and L have been given in the manuscript for the first time or where appropriate, see lines 155-156 on page 4 and line 192 on page 5 in the revised manuscript for details.

Point 5: What represent the different color in Fig. 13?

Response 5: The color of the curves in each subgraph in Fig. 13 has no special meaning, and the color of the curve between each two blue curves is changed only for the reader to more clearly distinguish the spectrum analysis results under each wind direction angle.

Point 6: What is "?" in the caption of Fig. 14. (a) ?

Response 6: The caption for the x-axis of Fig. 14(a) has been modified. The "?" that appears may be due to the display error when the word document uploaded during submission is converted into the PDF review file. We will submit a revised manuscript in PDF format to avoid this problem.

Point 7: Fig. 25-27: Please add what represents the colors in the plot or provide more information on the caption.

Response 7: The vorticity contours in the z-direction are shown in Figs. 25 to 27. The red color in the figures represents the positive vorticity of counterclockwise rotation, and the blue color represents the negative vorticity of clockwise rotation. These figures are similar to Fig. 5. Therefore, the above description has been added into the text corresponding to figure 5, see lines 233–236 on page 6 and page7 of the revised manuscript for details.

In addition, the legend of vorticity value is supplemented in Figs. 25, 27 and 28 of the revised manuscript.

Reviewer 2 Report

In this drafted manuscript, the authors have presented an investigation a numerical study related to moving surface boundary layer control on the wake of flow around a square cylinder. After the detailed glance, the main issues for suggestion can be seen as the following comments:

  • How did the authors determine the distances of computational domain such as 10L, 30 L and 10L? Are these distances sufficient for the development of the flow? Is there any citation?
  • The authors should show around the region of rotating cylinders as close-up view in terms of better understanding.
  • The authors need to ensure more information about mesh structure. Is the +y value taken into account?
  • The resolutions of few figures are not readable and too small. They must be reedited. Furthermore, there is a problem in x axis of Figure 14(a).
  • What is the novelty of paper? What’s the new? At least, it should be emphasized at the end of the introduction and conclusion part of paper.

In addition to comments mentioned above, the authors are suggested to add the current and actual studies including low Reynolds number aerodynamics and passive flow control studies as the following examples:

-     Investigation of pre-stall flow control on wind turbine blade airfoil using roughness element.

-     Identification of flow phenomena over NACA 4412 wind turbine airfoil at low Reynolds numbers and role of laminar separation bubble on flow evolution

- Experimental study of the wind turbine airfoil with the local flexibility at different locations for more energy output

-  Mapping of laminar separation bubble and bubble-induced vibrations over a turbine blade at low Reynolds numbers

-  Effect of partial flexibility over both upper and lower surfaces to flow over wind turbine airfoil

-  Traditional and New Types of Passive Flow Control Techniques to Pave the Way for High Maneuverability and Low Structural Weight for UAVs and MAVs

-  Electricity production from piezoelectric patches mounted over flexible membrane wing at low Reynolds numbers

Overall, comments and suggestions mentioned above should be taken into consideration carefully for better and scientific paper, allowing the researchers who interested in passive control methods in low Reynolds number.

Author Response

Response to Reviewer 2 Comments

applsci-1472032, Applied Sciences

Title: Moving surface boundary-layer control on the wake of flow around a square cylinder

Authors: Te Song, Xin Liu, Feng Xu

The insightful comments and helpful suggestions given by the reviewers are greatly appreciated. In response to the reviewers’ comments, this document is prepared and submitted together with the revised manuscript.

In the context below, the words in black are taken from the reviewer’s comments. Our responses are given in the sections that follow in red color, and the modifications in the revised manuscript according to the reviewer’s comments are marked in red color.

~~~~~~~~~~~~~~~~~~~~~~~~~~~~~~~~~~~~~~~~~~~~~~~~~~~

Reviewers' comments and Responses:

Reviewer #2:

In this drafted manuscript, the authors have presented an investigation a numerical study related to moving surface boundary layer control on the wake of flow around a square cylinder. After the detailed glance, the main issues for suggestion can be seen as the following comments:

Point 1: How did the authors determine the distances of computational domain such as 10L, 30 L and 10L? Are these distances sufficient for the development of the flow? Is there any citation? 

Response 1: The height of the computational domain in this study is 20L (L is the side length of the square cylinder), and the blockage ratio is 5% (L/20L), which meets the requirement of blockage ratio ≤ 5 in CFD numerical simulation. The downstream length of the computational domain is taken as 30L, which can ensure the full development of the downstream flow of the square cylinder. The accuracy of the flow field calculation can be seen from the comparison and verification results shown in Table 2. The determination of the computational domain of present study refers to the existing research results:

1)Sohankar et al. [39], H=20L,Xu=10L,Xd=3~26L;

2)Jan and Sheu [42], H=12L,Xu=6L,Xd=20L;

3)Cheng et al. [40], H=24L,Xu=12L,Xd=30L;

4)Abograis and Alshayji [43], H=20L,Xu=10L,Xd=15L;

5)Present,H=20L,Xu=10L,Xd=30L。

where H is the height of the computational domain, Xu is the distance from the model to the inlet, Xd is the distance from the model to the outlet.

The selection basis of the computational domain size has been added to the revised manuscript, see lines 158~162 on page 4 for details.

Point 2: The authors should show around the region of rotating cylinders as close-up view in terms of better understanding.

Response 2: This is a good suggestion. The diagrams of velocity contours and streamlines in the area around the rotating cylinders and near wake of the square cylinder have been added, see Fig. 26 on page 30 , and the corresponding description has been added in the main text (lines 606-641) of the revised manuscript.

Point 3: The authors need to ensure more information about mesh structure. Is the +y value taken into account?

Response 3: The more detailed information about mesh structure has been added to Section 2.3, see lines 213-216, 227-229 and red words in Table 1 in the revised manuscript.

The calculation in this paper is carried out at a low Reynolds of 200, using the laminar flow model without introducing the turbulent model, so the y+ value does not need to be considered. However, in this study, the grid independence analysis shown in Table 1 and the result verification shown in Table 2 have been given, which can indicate that the number of grids used in this calculation is sufficient to obtain the accurate flow field around the square cylinder.

Point 4: The resolutions of few figures are not readable and too small. They must be reedited. Furthermore, there is a problem in x axis of Figure 14(a).

Response 4: According to the reviewer's comment, in order to make the readers see more clearly, the scatter size, linetype and color of the following figures have been modified: Fig.4, Fig.6~10, and Fig.14~17.

The caption for the x-axis of Fig. 14(a) has been modified. The "?" that appears may be due to the display error when the word document uploaded during submission is converted into the PDF review file. We will submit a revised manuscript in PDF format to avoid this problem.

Point 5: What is the novelty of paper? What’s the new? At least, it should be emphasized at the end of the introduction and conclusion part of paper.

Response 5: This is a good comment. The new results are obtained through the present numerical study, such as the ones in Fig. 28 (The wake vorticity contour of the square cylinder under co-rotation mode), which is meaningful to the readers.

The corresponding application prospect is described in lines 670-675 with red color on page 31 in the revised manuscript.

In addition to comments mentioned above, the authors are suggested to add the current and actual studies including low Reynolds number aerodynamics and passive flow control studies as the following examples:

-    Investigation of pre-stall flow control on wind turbine blade airfoil using roughness element.

-    Identification of flow phenomena over NACA 4412 wind turbine airfoil at low Reynolds numbers and role of laminar separation bubble on flow evolution

-  Experimental study of the wind turbine airfoil with the local flexibility at different locations for more energy output

-  Mapping of laminar separation bubble and bubble-induced vibrations over a turbine blade at low Reynolds numbers

-  Effect of partial flexibility over both upper and lower surfaces to flow over wind turbine airfoil

-  Traditional and New Types of Passive Flow Control Techniques to Pave the Way for High Maneuverability and Low Structural Weight for UAVs and MAVs

-  Electricity production from piezoelectric patches mounted over flexible membrane wing at low Reynolds numbers

Overall, comments and suggestions mentioned above should be taken into consideration carefully for better and scientific paper, allowing the researchers who interested in passive control methods in low Reynolds number.

Response: In addition, the references suggested by the reviewer have been added to the ‘Introduction’ section, see lines 63-72 on page 2 in the revised manuscript for details, and the corresponding references have been added to the list of references.

Reviewer 3 Report

Very nice work.

1) P. 1 line 32 better use lessen instead of less 

2) You should explain what VIV stands for.

3) I would suggest to introduce a picture/sketch of the distribution of the measuring points.

4) Fig.18. It is not clear what U8 represents in the abscissa label.

Author Response

Response to Reviewer 3 Comments

applsci-1472032, Applied Sciences

Title: Moving surface boundary-layer control on the wake of flow around a square cylinder

Authors: Te Song, Xin Liu, Feng Xu

The insightful comments and helpful suggestions given by the reviewers are greatly appreciated. In response to the reviewers’ comments, this document is prepared and submitted together with the revised manuscript.

In the context below, the words in black are taken from the reviewer’s comments. Our responses are given in the sections that follow in red color, and the modifications in the revised manuscript according to the reviewer’s comments are marked in red color.

~~~~~~~~~~~~~~~~~~~~~~~~~~~~~~~~~~~~~~~~~~~~~~~~~~~

Reviewers' comments and Responses:

Reviewer #3:

Very nice work.

Point 1: P. 1 line 32 better use lessen instead of less.

Response 1: According to the reviewer's comment, “less” in the text has been replaced by “lessen”, as shown in line 32 with red color on page1 in the revised manuscript.

Point 2: You should explain what VIV stands for.

Response 2:  ‘VIV’ is the abbreviation for the Vortex Induced Vibration. According to the reviewer's comment, the meaning of VIV has been added to the text, see the red words in line 41 on page 1 for details in the revised manuscript.

Point 3: I would suggest to introduce a picture/sketch of the distribution of the measuring points.

Response 3: The detailed arrangement of the pressure measuring points is shown in Figure 3. A total of 32 measuring points are arranged on the surfaces of the model, among which 18 measuring points are evenly arranged on the straight edges of the square cylinder, and 7 measuring points are evenly arranged on the quarter arc of each rotating small cylinder. The main purpose is to obtain the pressure distribution characteristics on each straight edge of the square cylinder surfaces and on the quarter arc tangent to surfaces of the square cylinder. The detailed description of the measuring points arrangement can be seen in lines 179-183 on page 5 in the revised manuscript.

It should be noted here that each measuring point must be arranged in the flow field near the wall boundary. In order to ensure the accuracy of the results, the distance between each measuring point and the wall boundary is 0.1 mm.

Point 4: Fig.18. It is not clear what U represents in the abscissa label.

Response 4: The caption of the abscissa for Fig. 18 represents the dimensionless time t*=tU/L, where U represents the uniform incoming wind velocity at the inlet boundary. The value of U (0.054 m/s) has been given in the revised manuscript, see line 192 on page 5 for details.

Round 2

Reviewer 2 Report

The authors have reedited the paper and responded to the questions and criticisms well. The authors do not need to minor or major touch to the drafted paper. The paper is publishable in Applied Sciences.